# STYLE EQUALIZATION: UNSUPERVISED LEARNING OF CONTROLLABLE GENERATIVE SEQUENCE MODELS

## ABSTRACT

Controllable generative sequence models with the capability to extract and replicate the style of specific examples enable many applications, including narrating audiobooks in different voices, auto-completing and auto-correcting written handwriting, and generating missing training samples for downstream recognition tasks. However, typical training algorithms for these controllable sequence generative models suffer from the training-inference mismatch, where the same sample is used as content and style input during training but different samples are given during inference. In this paper, we tackle the training-inference mismatch encountered during unsupervised learning of controllable generative sequence models. By introducing a style transformation module that we call *style equalization*, we enable training using different content and style samples and thereby mitigate the training-inference mismatch. To demonstrate its generality, we applied style equalization to text-to-speech and text-to-handwriting synthesis on three datasets. Our models achieve state-of-the-art style replication with a similar mean style opinion score as the real data. Moreover, the proposed method enables style interpolation between sequences and generates novel styles.

## 1 INTRODUCTION

The goal of controllable generative sequence models is to generate sequences containing target content in a target style. With the capability to select speaker voices, multi-speaker text-to-speech models have been successfully adopted in many voice assistants (Gibiansky et al., 2017; Ping et al., 2018; Hayashi et al., 2020). Many applications, however, require style controllability beyond selecting speaker voices. For example, to perfectly reconstruct a speech example, we need to replicate not only the speaker's voice characteristics but also all aspects of style about the sample, including but not limited to prosody, intonation dynamics, background noise, echo, and microphone response appeared in the given sample. To analyze failures or biases of a downstream recognizer, we need a style representation that models the entire style distribution, beyond speaker identity. In these applications, style represents all information (except the content) to exactly reconstruct a sample, as illustrated in Fig. 1a. Notice that to represent the time-dependent information of a sample, style is itself a sequence and changes over time, instead of a fixed vector. Moreover, even when the same speaker utters the same content, the resulting audios can contain different styles. To capture the large variation, the style representation should be learned in an *unsupervised* manner from a reference sample, rather than using a few human-annotated attributes.

Our goal is to learn a controllable generative sequence model that controls its style with a reference example (*e.g.*, an existing audio) and controls the content with a content sequence (*e.g.*, text), as shown in Fig. 1b. Our training dataset $\mathcal{X}$ is composed of $\left\{ (\mathbf{x}^i, \mathbf{c}^i) \right\}_{i=1,\dots,n}$, where $\mathbf{x}^i = \left[ x_1^i \ \dots \ x_{T_i}^i \,|\, x_t^i \in \mathbb{R}^d \right]$ is the i-th sample and $\mathbf{c}^i = \left[ c_1^i \ \dots \ c_{N_i}^i \,|\, c_j^i \in \mathbb{R}^m \right]$ is the corresponding content sequence. Note that in general, $\mathbf{x}^i$ and $\mathbf{c}^i$ have different lengths, *i.e.*, $T_i \neq N_i$, and we do not have the alignment between them. For example, in text-to-speech synthesis, $\mathbf{x}^i$ is the mel-spectrogram of an audio sample, $\mathbf{c}^i$ is the corresponding phonemes of the spoken words, and we do not have the mapping between the phonemes and mel-spectrogram. We also do not have any style supervision, including speaker or attribute labels, nor any grouping of the data based on style.

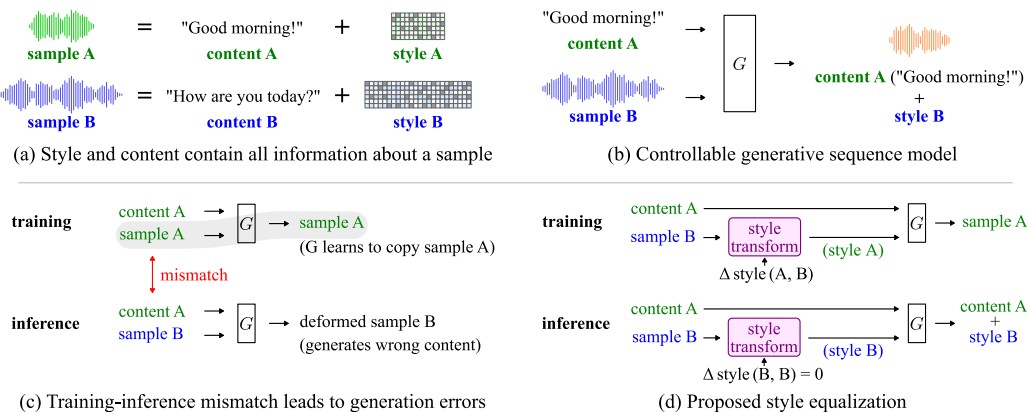

(a) Style and content contain all information about a sample

(b) Controllable generative sequence model

(c) Training-inference mismatch leads to generation errors

(d) Proposed style equalization

Figure 1: **Controllable generative models with sample-level style control.** (a) The information contained in a sample can be divided into content (*i.e.*, the text) and style (*i.e.*, all other information besides content). (b) Our goal during inference is to generate samples containing target content A in the style of sample B. Notice that sample B generally contains a different content. (c) There exists a training-inference mismatch when learning these models in typical unsupervised training of controllable generative models. During training, the *same* sample is used as content input and style input, whereas during inference, content and style inputs are from different samples, *i.e.*, the reference style sample contains a different content than the target content. The mismatch leads to incorrect content generation during inference. (d) To mitigate the training-inference mismatch, the proposed style equalization takes unpaired samples as input during both training and inference. It transforms the style of sample B to that of A by estimating their style difference.

While the unsupervised setting requires only the essential information (*i.e.*, samples and their content), it makes learning a controllable generative sequence model very challenging. The main challenge is the mismatch between the inputs used during training and inference. As shown in Fig. 1c, during inference we pair arbitrary content A and reference sample B as inputs. However, due to the lack of ground truth containing content A and in the style of B, during training we pair content A and sample A. In other words, we train the model under the *parallel setting* where the reference style input contains the input content, but we use the model in the *non-parallel setting* (where the reference style contains a different content than the target content) during inference. Due to the training-inference mismatch, a well-performing model during training may perform poorly during inference. If a generative model learns to utilize the content information in the style example, during inference the generative model will generate wrong content. This phenomenon is called *content leakage* (Hu et al., 2020). In an extreme case, a model can learn to copy the reference sample to the output; despite its perfect training loss, it is useless because it always generates wrong content in practice.

This paper proposes a simple but effective technique to deal with the training-inference mismatch when we learn controllable auto-regressive models in an unsupervised manner. As shown in Fig. 1d, we train the model under the non-parallel setting, *i.e.*, we pair arbitrary content A with an arbitrary sample B from the training dataset. Instead of directly using sample B as style (in which case we have no ground truth), we jointly learn a style transformation function, which estimates the style difference between A and B and transforms the style of sample B to the style of A. The generative model then takes content A and the transformation output (that contains the style of A) to reconstruct sample A. The proposed method enables us to use sample A as the ground truth while learning in the non-parallel setting—the intended usage during inference. Additionally, our method provides a systematic way to interpolate between the style of two samples by scaling the estimated style difference between two reference samples. We call the method *style equalization*. Note that for style equalization to work, the style transformation and difference estimator need to be carefully designed, such that no content information from content A can be transferred through sample B. We defer the discussion to Sec. 4. NEW

The proposed method is general and can be applied to different sequence signals. We apply the proposed method on two signal domains, speech and online-handwriting, and evaluate the performance carefully via quantitative evaluation (by computing content error rates) and conducting qualitative user studies. Experimental results show that our method outperforms various unsupervised controllable FIX

sequence generative models, even when they have additional style supervision like speaker labels. On LibriTTS, style equalization achieves close style replication (3.5 real oracle vs. 3.5 proposed in style opinion score) and content reproduction errors (6.6% real oracle vs. 9.5% proposed) to real samples.

FIX

## 2 RELATED WORK

Controllable generative sequence models are not new in the literature; however, the majority of these methods require style supervision, whereas the paper develops an unsupervised-style method. Table 6 provides an overview of the related works.

NEW

**Unsupervised-style sequence models.** Unsupervised methods extract style information directly from samples, *i.e.*, without any style labels or pretrained style embeddings. Existing unsupervised methods train models under the parallel setting, as shown in Fig. 1c. To prevent content leakage, most existing methods introduce a bottleneck on the capacity of the style encoder by representing style as a single (time-invariant) vector and limiting its dimension (Wang et al., 2018; Hsu et al., 2018; Hu et al., 2020; Ma et al., 2018). Wang et al. (2018) propose Global Style Token (GST), which represents a style vector as a linear combination of a learned dictionary (called style tokens) shared across the dataset. The number of style tokens (the implicit dimension of the style vector) is carefully controlled to prevent content leakage. As we will see in Sec. 3, the bottleneck not only reduces the amount of content information contained in the style vector but also sacrifices style information.

NEW

Alternative loss formulations have also been proposed to limit content information contained in the style representation. Hu et al. (2020) minimize the mutual information between the style vector and the content sequence but requires a pretrained content encoder and adversarial learning, which makes training their model difficult. Hsu et al. (2018) approximate the posterior distribution of the style vector using a mixture of Gaussian distributions with a small number of mixtures. Ma et al. (2018) utilize a discriminator conditioned on both the generated output and the content (similar to a content recognizer). Akuzawa et al. (2018) anneal the Kullback-Leibler divergence to control the amount of information contained in style. Henter et al. (2018) utilize phoneme segmentation (McAuliffe et al., 2017) to avoid learning the alignment between content **c** and output **x**.

NEW
NEW
NEW

Priming is a technique that is introduced to control the style of auto-regressive generative sequence models (Graves, 2013; Aksan et al., 2018). Since the hidden state of a Recurrent Neural Network (RNN) contain all information about current generation, including style, we can initialize the RNN by pre-rolling the reference sample through the RNN. Utilizing priming requires the content of the reference style. For example, Aksan et al. (2018) learn a character recognizer and use it during inference. Moreover, since the hidden state contains residual content from the reference example, it often generates unexpected artifacts at the beginning of the sequence, as will be seen in Sec. 5.

NEW

**Supervised-style sequence methods.** Many existing controllable generative models require style supervision, either directly by passing attribute labels as inputs or implicitly by grouping training data with their attribute labels. In the following, we briefly introduce various supervised controllable sequence models. While using style supervision avoids training-inference mismatch, it limits the style control over a few sparsely-defined attribute classes. For instance, given a speech audio, we can recognize the spoken texts, the accent, or even the speaker, but provided solely with these attribute labels, it is impossible to exactly reconstruct the original speech audio. The sparsely-defined attributes are insufficient to capture the entire style information.

User identifications or their embeddings have been used to learn multi-speaker text-to-speech models (Jia et al., 2018; Gibiansky et al., 2017; Kameoka et al., 2020; Donahue et al., 2020; Chen et al., 2021; Dhariwal et al., 2020; Valle et al., 2020; Kim et al., 2020; Hayashi et al., 2020; Sun et al., 2020),voice conversion models (Qian et al., 2019) and handwriting models (Kotani et al., 2020; Bhunia et al., 2021; Kang et al., 2020; Davis et al., 2020). In addition to user identifications, predefined features like pitch, phoneme duration, loudness, and timbre have also been used by existing methods (Ren et al., 2020; Qian et al., 2020; Dhariwal et al., 2020; Valle et al., 2020). Instead of using speaker labels as input, Kameoka et al. (2018); Kaneko and Kameoka (2018); Kaneko et al. (2019a;b) group training samples by their speaker labels and apply adversarial learning to learn voice conversion models that change speaker voices while keeping the content of the input.

NEW

**Image methods.** Controllable generative models have also been developed for images (Härkönen et al., 2020; Esser et al., 2019; Singh et al., 2019; Lample et al., 2017; Karras et al., 2020; Brock et al., 2019; Collins et al., 2020; Shen et al., 2020; Esser et al., 2020; Goetschalckx et al., 2019; Pavllo et al., 2020; Zhang et al., 2018), which control the object class, pose, lighting, *etc.*, of an image. Many image style transform methods have also been developed (Isola et al., 2017; Zhu et al., 2017; Gatys et al., 2016). However, there is a fundamental difference between image and sequence problems. In image generative models, we do not need to learn the content-output alignment. The content is usually defined globally as an image class or as pixel labels, *e.g.*, segmentation map. In contrast, our content is given as text, the output is mel-spectrogram of a waveform, and the content and output have different lengths. To utilize the input content sequence, generative sequence models need to align the content and the output sequences and translate text to the output signal modality. The complication exacerbates the training-inference mismatch for sequence methods, since copying the style input is easier than utilizing the input content.

## 3 CONTROLLABLE GENERATIVE SEQUENCE MODELS

We focus on learning controllable auto-regressive generative models, $p(x_t|z_t, \mathbf{x}_{1..t-1}, \mathbf{c})$, where $\mathbf{x} = [x_1, \ldots, x_T]$ is the output sequence, $\mathbf{c}$ is the content sequence, and $\mathbf{z} = [z_1, \ldots, z_T \mid z_t \in \mathbb{R}^\ell]$ is the reference style information. Note that, in our model, style is also represented as a sequence that changes over time. Under the style-unsupervised setting, we are given a dataset $\mathcal{X} = \{(\mathbf{x}^i, \mathbf{c}^i), i \in \{1 \ldots n\}\}$ that contains the ground-truth output sequence $\mathbf{x}^i$ and the corresponding content $\mathbf{c}^i$, but we do not have style supervision on $\mathbf{z}$. Therefore, we treat $z_t$ as a latent variable with a learnable prior distribution $p(z_t|\mathbf{x}_{1..t-1}, \mathbf{c})$ and optimize the log-likelihood of $\mathbf{x}$ conditioned on $\mathbf{c}$, $\mathbb{E}_{(\mathbf{x},\mathbf{c})} \log p(\mathbf{x}|\mathbf{c})$. Specifically, we maximize a variational lower bound of the likelihood as VRNN (Chung et al., 2015): NEW

$$\mathbb{E}_{(\mathbf{x},\mathbf{c})} \log p(\mathbf{x}|\mathbf{c}) = \mathbb{E}_{(\mathbf{x},\mathbf{c})} \sum_{t=1}^{T} \log p(x_t|\mathbf{x}_{1..t-1}, \mathbf{c}) = \mathbb{E}_{(\mathbf{x},\mathbf{c})} \sum_{t=1}^{T} \log \mathbb{E}_{z_t \sim p(z_t|\mathbf{x}_{1..t-1},\mathbf{c})} p(x_t|z_t, \mathbf{x}_{1..t-1}, \mathbf{c})$$

$$\geq \mathbb{E}_{(\mathbf{x},\mathbf{c})} \sum_{t=1}^{T} \mathbb{E}_{z_t \sim q(z_t|\mathbf{x},\mathbf{c})} \log p(x_t|z_t, \mathbf{x}_{1..t-1}, \mathbf{c}) - \mathcal{D}_{KL}\left(q(z_t|\mathbf{x}, \mathbf{c}) \,\|\, p(z_t|\mathbf{x}_{1..t-1}, \mathbf{c})\right), \quad (1)$$

where $\mathcal{D}_{KL}$ represents the Kullback-Leibler (KL)-divergence. In eq. (1), we use the chain rule to expand $p(\mathbf{x}|\mathbf{c})$ into $p(x_1|\mathbf{c}) \, p(x_2|x_1, \mathbf{c}) \cdots p(x_T|\mathbf{x}_{1..T-1}, \mathbf{c})$, introduce the variational approximation $q(z_t|\mathbf{x}, \mathbf{c})$ of the posterior distribution $p(z_t|\mathbf{x}, \mathbf{c})$ for all $t$, and apply Jensen's inequality. Note that since $q$ is a variation approximation of the posterior distribution, it can be conditioned on any variable.

Fig. 2a shows an overview of the network used in the paper — the input content $\mathbf{c}$ is processed by the content attention, the style encoder (shown in green) models $q(z_t|\mathbf{x}^r, \mathbf{c})$, and the decoder (shown in gray) models $p(x_t|z_t, \mathbf{x}_{1..t-1}, \mathbf{c})$ using output from the style encoder and the content attention. Note that during inference, $\mathbf{x}^r$ is the reference example that we replicate the style of, and it generally contains an unrelated content, *i.e.*, $\mathbf{c}^r \neq \mathbf{c}$. During training, the ground-truth sample $\mathbf{x}$ is used as the reference style to optimize eq. (1), *i.e.*, $\mathbf{x}^r = \mathbf{x}$ and $\mathbf{c}^r = \mathbf{c}$, which is different from inference. Therefore a generative model can learn to copy the style input to the output (and ignore the content input), leading to incorrect generation during inference. This phenomenon can be remedied by limiting the capacity of the style encoder, *e.g.*, by decreasing the dimensionality of the style representation. However, to achieve the lower bound of eq. (1) and hence a higher generation quality, we need the style encoder to contain enough capacity such that $q(z_t|\mathbf{x}, \mathbf{c}) \equiv p(z_t|\mathbf{x}, \mathbf{c})$ for all $t$. In this paper, we provide an alternative training procedure that bypasses this trade-off — we use a high-capacity style encoder (shown in Fig. 2b) and prevent content leakage.

## 4 STYLE EQUALIZATION

Let $(\mathbf{x}, \mathbf{c})$ and $(\mathbf{x}', \mathbf{c}')$ be two samples from the training set. To match the inference setting, we should train the model using $\mathbf{c}$ as content and $\mathbf{x}'$ as style input. However, neither $\mathbf{x}$ nor $\mathbf{x}'$ can be used as the ground-truth output sequence. If we use $\mathbf{x}$ as the ground-truth output sequence but $\mathbf{x}'$ as style input, the generative model will learn to ignore the style encoder since $\mathbf{x}'$ contains unrelated style information. In other words, the variational approximation $q(\mathbf{z}|\mathbf{x}', \mathbf{c})$ is a poor approximation to the true posterior $p(\mathbf{z}|\mathbf{x}, \mathbf{c})$. Alternatively, if we use $\mathbf{x}'$ as the ground-truth output sequence, the content given in $\mathbf{c}$ will be ignored.

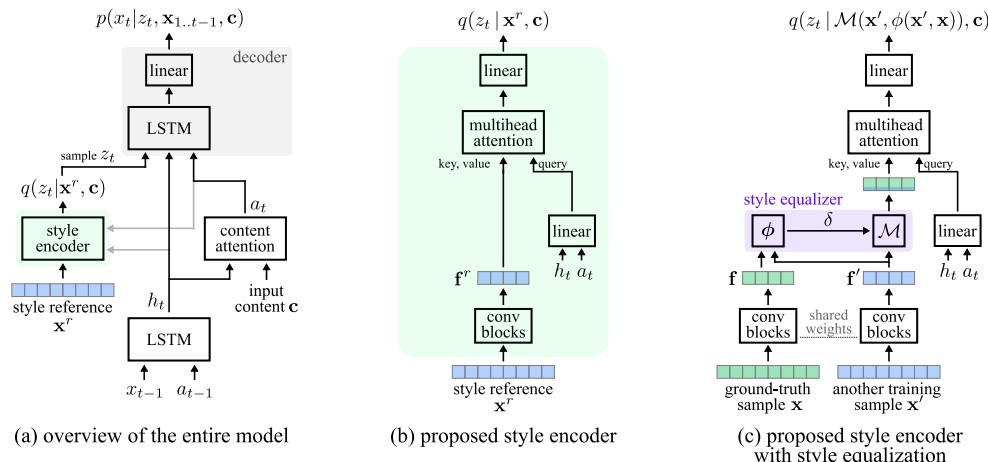

(a) overview of the entire model    (b) proposed style encoder    (c) proposed style encoder with style equalization

Figure 2: **Model used in the paper.** (a) an overview of the entire model, which includes a style encoder (in green), content attention, and decoder (in gray). Note that the input content $\mathbf{c}$ can be the output of a content-embedding network (used in speech synthesis) or one-hot encoding of characters (used in handwriting synthesis). $a_t$ is the output of the content attention at time $t$, which is a linear combination of the elements in $\mathbf{c}$. (b) the proposed style encoder without style equalization. (c) the proposed style encoder with the style equalization. $\phi$ computes the vector $\delta$ that encodes the style difference between $\mathbf{x}'$ and $\mathbf{x}$. $\mathcal{M}$ applies this transformation to $\mathbf{x}'$ to match the style of $\mathbf{x}$.

We introduce a learnable style transformation function $\mathcal{M}(\mathbf{x}', \delta)$ that we use to transform the style of $\mathbf{x}'$ by the amount specified by a vector $\delta \in \mathbb{R}^k$. Instead of directly using $\mathbf{x}'$ as the style input, we first estimate the style difference vector between $\mathbf{x}$ and $\mathbf{x}'$ with a learnable function $\phi$, *i.e.* $\delta = \phi(\mathbf{x}', \mathbf{x})$; we then transform $\mathbf{x}'$ with $\mathcal{M}(\mathbf{x}', \delta)$. By jointly optimizing $\mathcal{M}$, $\phi$, and the rest of the generative model using eq. (1), the model learns to transfer style information from $\mathbf{x}$ through $\mathcal{M}$ to maximize the log-likelihood of the ground-truth $\mathbf{x}$. In other words, we approximate the posterior distribution $p(\mathbf{z}|\mathbf{x}, \mathbf{c})$ with $q(\mathbf{z}|\mathcal{M}(\mathbf{x}', \phi(\mathbf{x}', \mathbf{x})), \mathbf{c})$, which is a better approximation than $q(\mathbf{z}|\mathbf{x}', \mathbf{c})$. We call this method *style equalization*. Note that, for style equalization to be successful, $\mathcal{M} \circ \phi$ should not transfer any content-related information (*e.g.*, copy the entire sequence) from $\mathbf{x}$ but only its style information so that the decoder will utilize the transferred style and will rely on provided content input to generate the output. Therefore the design of $\mathcal{M}$ is critical.

**Design of $\mathcal{M}$ and $\phi$.** An important observation we use in the design of $\mathcal{M}$ and $\phi$ is that content information (*i.e.*, sequence of phonemes or characters) is strongly time-dependent whereas the style can be reasonably well approximated by a time-independent representation (*e.g.*, voice characteristics of a speaker and microphone response, *etc.*). By designing the style difference estimator $\phi$ such that no sequence information is stored in the difference vector $\delta$, we can satisfy that content-related information is not leaked, but the function can still transfer the time-independent style information. We achieve this using convolutional filters and average pooling over the time dimension.

As shown in Fig. 2c, to estimate the style difference vector between two sequences $\mathbf{x}$ and $\mathbf{x}'$, we first compute their style features $\mathbf{f}$ and $\mathbf{f}'$ using a convolutional network. Note that $\mathbf{f}$ and $\mathbf{f}'$ are $s$-dimensional feature sequences with different lengths. We define

$$\phi(\mathbf{x}', \mathbf{x}) = \mathrm{avg}(A\,\mathbf{f}) - \mathrm{avg}(A\,\mathbf{f}') \quad \text{and} \quad \mathcal{M}(\mathbf{x}', \phi(\mathbf{x}', \mathbf{x})) = \mathbf{f}' + A^\top \phi(\mathbf{x}', \mathbf{x}), \quad (2)$$

where avg represents taking mean across time, $A \in \mathbb{R}^{k \times s}$ is a learnable linear transform, and $\mathbf{f}' + A^\top \phi(\mathbf{x}', \mathbf{x})$ means that the vector $A^\top \phi(\mathbf{x}', \mathbf{x}) \in \mathbb{R}^s$ is added to each time step of $\mathbf{f}'$. Intuitively, the design assumes that the style information lies on a $k$-dimensional subspace, and we equalize the style between $\mathbf{x}'$ and $\mathbf{x}$ by minimizing their differences in the subspace. It also satisfies the identity property — $\phi(\mathbf{x}^r, \mathbf{x}^r) = 0$ and $\mathcal{M}(\mathbf{x}^r, 0) = \mathbf{f}^r$ — which allows us to treat style equalization as a training procedure and remove it from the model during inference (as shown in Fig. 2b).

**Interpolation between two styles.** Once learned, $\mathcal{M}$ and $\phi$ can be used to manipulate style during inference. Given two style references, $\mathbf{x}^s$ and $\mathbf{x}^t$, we interpolate between them with $\mathcal{M}(\mathbf{x}^s, \alpha\,\phi(\mathbf{x}^s, \mathbf{x}^t))$, where a scalar $\alpha \in \mathbb{R}$ controls the interpolation. By changing $\alpha$, we traverse

a one-dimensional manifold that starts from the original style (with $\alpha = 0$) and ends at the target style (with $\alpha = 1$). Note that, unlike existing generative models that support style interpolation in post-processing, $\mathcal{M}$ and $\phi$ are trained to transform style by design.

## 5 EXPERIMENTS

To demonstrate the generality of the proposed method, we train and evaluate it on two different signals, speech and handwriting, with the same model architecture design. In the following, we will introduce the model architecture used in the experiments, the baselines, the metrics, and the results. More details are provided in the supplemental material.

NEW

### 5.1 MODEL ARCHITECTURE

Our model is auto-regressive and composed of (i) a decoder that is modeling $p(x_t|z_t, \mathbf{x}_{1..t-1}, \mathbf{c})$, (ii) a content attention module, (iii) a style encoder that is modeling $q(z_t|\cdot, \mathbf{c})$, and (iv) a network that models the prior distribution of $z_t$. Fig. 2a shows an overview of the model. The backbone of the model, namely the content attention and the decoder, uses a standard architecture that was proposed by Graves (2013) for handwriting synthesis and later extended to speech in variations of Tacotron (Shen et al., 2018; Wang et al., 2018). The variational approximation $q(z_t|\cdot)$ and the prior distribution are modeled as multivariate Gaussian distributions with a diagonal covariance matrix.

Our style encoder is composed of a convolutional network and a multi-head attention layer, as shown in Fig. 2b. The convolutional network extracts the style feature sequence $\mathbf{f}^r$ from the reference style input $\mathbf{x}^r$. We use multi-head attention to extract relevant style information at every time step from $\mathbf{f}^r$ with the query computed from the hidden state of the LSTM, $h_t$, which contains information about past generations, and the currently focused content $a_t$. Thus, our style representation is a time-varying sequence. The intuition is that if the model utters a particular phoneme, it should be able to find the information in the style reference and mimic it.

For style equalization, we insert $\mathcal{M}$ and $\phi$ into the style encoder, as shown in Fig. 2c. Since style equalization is only able to transfer time-independent information, when we utilize this procedure, the network will not be able to learn time-dependent style. To enable learning time-dependent style information during training, half of the batches, we use $\mathbf{x}' = \mathbf{x}$, which means that the difference vector $\delta = 0$, hence the decoder directly uses the ground-truth style information which contains time-dependent style information. We analyze the effect of style attention and its ability to represent time-varying style information in Sec. 5.4.

### 5.2 SPEECH SYNTHESIS

We train and evaluate the proposed method on two multi-speaker speech datasets. VCTK dataset (Yamagishi et al., 2019) contains 110 speakers and 44 hours of speech, and LibriTTS dataset (Zen et al., 2019) contains 2,311 speakers and 555 hours of speech in the training set.[1]

NEW

**Baselines.** We compare the proposed method with Global Style Tokens (GST-n) (Wang et al., 2018) with various numbers of tokens n. For the sake of completeness, we also compare with Tacotron 2 (Shen et al., 2018) (even though it does not have style control), Tacotron-S and GST-nS. Tacotron-S and GST-nS are Tacotron and GST-n with pretrained style embedding (Snyder et al., 2018) that was trained on the VoxCeleb dataset (Chung et al., 2018), respectively. The VoxCeleb dataset contains $2,000$ hours of speech from $7,000$ speakers and has a large variation in recording conditions. We use ESPnet-TTS (Hayashi et al., 2020), a widely used implementation of the baselines and follow their training recipe. All the baselines are trained using the same dataset as the proposed model[2]. They achieve similar performance as those listed in the original papers. All the methods output 80-dimensional mel-spectrograms with the sampling rate equal to 22,050 Hz and the window size equal to 1,024, which are converted to waveforms using a pretrained WaveGlow vocoder (Prenger et al., 2019). The content input is represented as phonemes.

NEW

---

[1] We combine train-clean-100, train-clean-360, and train-other-500.

[2] Tacotron-S and GST-nS use additional VoxCeleb dataset for style supervision

Table 1: **Quantitative results on VCTK dataset.** The reference style inputs are seen (randomly selected from the training set). WER measures content accuracy; cosine-similarity (cos-sim) and avgRank measure style similarity.

| Method | Parallel text | | | Nonparallel text | | |
|---|---|---|---|---|---|---|
| | WER (%) | cos-sim ↑ | avgRank ↓ | WER (%) | cos-sim ↑ | avgRank ↓ |
| Tacotron | 16.0 ± 1.7 | 0.05 ± 0.13 | 53.1 ± 29.1 | 16.4 ± 1.2 | 0.05 ± 0.12 | 53.9 ± 27.8 |
| Tacotron-S | 13.6 ± 0.7 | 0.24 ± 0.18 | 16.4 ± 20.9 | 16.3 ± 0.4 | 0.22 ± 0.18 | 18.0 ± 21.9 |
| GST-16 | 18.6 ± 0.9 | 0.23 ± 0.15 | 21.4 ± 21.9 | 18.5 ± 1.1 | 0.23 ± 0.16 | 21.1 ± 22.4 |
| GST-64 | 16.9 ± 0.5 | 0.23 ± 0.17 | 24.4 ± 23.1 | 27.5 ± 0.4 | 0.22 ± 0.16 | 25.2 ± 24.4 |
| GST-16S | 8.3 ± 0.1 | 0.34 ± 0.18 | 10.8 ± 15.2 | 17.7 ± 0.8 | 0.31 ± 0.17 | 13.0 ± 20.0 |
| GST-64S | 14.1 ± 0.3 | 0.33 ± 0.18 | 11.4 ± 16.3 | 24.7 ± 1.0 | 0.32 ± 0.18 | 12.7 ± 18.1 |
| Proposed | **7.4 ± 0.2** | **0.73 ± 0.12** | **1.5 ± 2.1** | **9.5 ± 0.4** | **0.64 ± 0.14** | **1.9 ± 4.2** |
| Oracle | 6.6 ± 0.0 | 1.0 ± 0.0 | 1.0 ± 0.0 | 6.6 ± 0.0 | 0.57 ± 0.16 | 1.6 ± 4.1 |

Table 2: Quantitative results on LibriTTS-all-960 dataset.

| Method | Seen speakers, parallel text | | | Seen speakers, nonparallel text | | | Unseen speakers, nonparallel text | | |
|---|---|---|---|---|---|---|---|---|---|
| | WER (%) | cos-sim ↑ | avgRank ↓ | WER (%) | cos-sim ↑ | avgRank ↓ | WER (%) | cos-sim ↑ | avgRank ↓ |
| Tacotron | 64.4 ± 4.1 | 0.00 ± 0.10 | 1218 ± 671 | 52.2 ± 1.7 | 0.01 ± 0.10 | 1140 ± 651 | 52.2 ± 0.6 | 0.00 ± 0.10 | 847 ± 563 |
| Tacotron-S | 14.9 ± 0.0 | 0.23 ± 0.22 | 430 ± 584 | 18.7 ± 0.2 | 0.24 ± 0.22 | 370 ± 562 | 13.5 ± 0.0 | 0.16 ± 0.16 | 289 ± 436 |
| GST-64 | 38.2 ± 2.2 | 0.12 ± 0.19 | 706 ± 701 | 33.3 ± 3.4 | 0.12 ± 0.19 | 700 ± 739 | 30.4 ± 2.2 | 0.09 ± 0.16 | 535 ± 610 |
| GST-192 | 19.3 ± 0.3 | 0.10 ± 0.17 | 786 ± 725 | 17.8 ± 0.6 | 0.09 ± 0.16 | 823 ± 719 | 18.0 ± 0.7 | 0.07 ± 0.14 | 587 ± 582 |
| GST-64S | 19.7 ± 0.7 | 0.39 ± 0.23 | 150 ± 305 | 20.4 ± 1.6 | 0.40 ± 0.23 | 143 ± 334 | 16.5 ± 0.2 | 0.28 ± 0.17 | 121 ± 259 |
| GST-192S | 13.8 ± 0.7 | 0.39 ± 0.23 | 137 ± 309 | 15.4 ± 1.1 | 0.41 ± 0.22 | 126 ± 317 | 13.4 ± 0.2 | 0.29 ± 0.18 | 139 ± 316 |
| Proposed | **6.2 ± 0.5** | **0.82 ± 0.14** | **1.7 ± 4.1** | **9.4 ± 0.3** | **0.78 ± 0.14** | **1.8 ± 6.0** | **7.6 ± 0.9** | **0.57 ± 0.15** | **7.4 ± 42.6** |
| Oracle | 6.5 ± 0.0 | 1.0 ± 0.0 | 1.0 ± 0.0 | 6.5 ± 0.0 | 0.85 ± 0.06 | 1.0 ± 0.0 | 6.5 ± 0.0 | 0.50 ± 0.23 | 3.6 ± 24.9 |

Table 3: Style opinion scores of speech synthesizers.

| VCTK, seen speakers | | | | LibriTTS, seen speakers | | | | LibriTTS, unseen speakers | | | |
|---|---|---|---|---|---|---|---|---|---|---|---|
| GST-64 | GST-16S | Proposed | Oracle | GST-192 | GST-192S | Proposed | Oracle | GST-192 | GST-192S | Proposed | Oracle |
| 2.1 ± 1.0 | 3.3 ± 0.9 | **3.8 ± 0.4** | 3.8 ± 0.4 | 1.4 ± 0.6 | 2.8 ± 1.0 | **3.6 ± 0.6** | 3.5 ± 0.9 | 1.2 ± 0.5 | 2.6 ± 0.9 | **3.5 ± 0.7** | 3.5 ± 0.9 |

**Metrics.** We measure the content generation errors as Word Error Rate (WER), using a pretrained speech recognition model, ESPnet (kamo naoyuki, 2021). To evaluate the style replication accuracy of the methods, we use a speaker classification network (Deng et al., 2019) and measure the style similarity between reference and output generations. We report `cos-sim`, which is the average cosine similarity between reference and output generations, and `avgRank`, which is the average rank of the reference speaker out of all speakers based on their cosine similarities. NEW

We also compute the style opinion score following the protocol used by Zhao et al. (2020). To evaluate the style similarity between a generated output and a style reference, users were given pairs of reference and synthesized audio, and asked if "the two samples could have been produced by the same speaker in a similar environmental condition", and asked to score with " 4 (Absolutely same)", "3 (Likely same)", "2 (Likely different)", "1 (Absolutely different)". We synthesized 100 samples using each method with the same style example and target content. A total of 15 users participated in the study, and we collected 630 responses in total.

We also provide an oracle (a pseudo upper-bound) where we select a different real speech sample from the same speaker from the dataset, and evaluate style similarity and content error. This provides a good calibration for our evaluation metrics and opinion studies.

**Results.** Table 1 shows the results on VCTK dataset. Let us first look at the parallel-text setting. Without any style control, Tacotron achieves a low cosine similarity. While GST and GST-nS improve cosine similarities, the proposed method achieves the highest similarity and lowest avgRank.

When comparing the WERs between the parallel and non-parallel settings, we can see the adversarial effect of content leakage — the models with a high-capacity style encoder (GST-64, GST-16S, and GST-64S) produce a much higher word-error rate in the non-parallel setting than in the parallel setting, while the small capacity GST-16 is largely unaffected. In comparison, the proposed method achieves a high cosine similarity and a similar word-error rate as the oracle, which demonstrates robustness to content leakage despite using the high-capacity style encoder.

Table 2 shows the results on LibriTTS dataset. As can be seen from the results, the large variety (*e.g.*, more accents, higher background noise, difficult microphone effects, etc) makes it a more difficult dataset than VCTK, as reflected by the high WERs of Tacotron. The proposed method, on the other hand, is able to learn from the noisy data and mimic the wide variety of styles in the dataset, including the voice characteristics and the background noise. In addition, since the training set contains more than $2,000$ speakers, the proposed method learns to generalize and mimic the voices of unseen speakers in the validation set, as demonstrated by the low WER and high cosine similarity in the right-most column of Table 2.

Table 3 shows the results of style opinion score evaluation on the two datasets. The proposed method achieves the highest score among the synthesizers. Furthermore, it achieves similar scores to the oracle. The result demonstrates the effectiveness of the proposed method in style replication. The method also enables sampling styles from the prior distribution and interpolating between two styles. Please see the supplemental material for these results.

## 5.3 ONLINE-HANDWRITING SYNTHESIS

Online-handwriting synthesis aims to generate sequences of pen movements on a writing surface. A handwriting sample is represented as a sequence of $(x, y, p)$ triplets, where $x$ and $y$ are the coordinates of the pen on the surface, and $p$ is a binary variable indicating whether the pen touches the surface over time. We apply the proposed method to a subset of a proprietary dataset collected for research and development. The subset consists of 600k online handwriting samples that were written by 1,500 people in English, French, German, Italian, Spanish, and Portuguese.

**Baselines and metrics.** We compare with the method proposed by Graves (2013) that uses priming for style encoding and an ablation of our model which uses the same style encoder but without style equalization. Note that all these models use the same decoder and content attention. Similar to speech, we measure content generation error as Character Error rate (CER) with a pretrained handwriting recognizer, and we conduct style-opinion-score study on 12 users and collected 320 responses. We also provide an oracle where we select a different real handwriting sample from the same writer from the dataset and compute the metrics.

**Results.** Fig. 3a shows synthesized handwriting from each of the methods on unseen style examples. As can be seen, while Graves (2013) with priming can replicate the reference style, it outputs artifacts at the beginning, and the style replication is worse than the proposed method. The model without style equalization produces high-quality replication in the parallel setting; however, it suffers severely from content leakage under the non-parallel setting and produces wrong content. In comparison, the proposed method generates correct content and replicate the style. We demonstrate the capability to interpolate between reference styles (*e.g.*, between cursive and printed style) in Fig. 3b. Our method also supports sampling styles from the learned prior distribution, as shown in Fig. 3c.[3]

Fig. 3d shows the CERs and style opinion scores for unseen style references. Due to the residual information in the hidden state, priming significantly increases the CERs for Graves (2013). Without style equalization, the model fails to synthesize legible handwriting in the nonparallel-text setting due to the content leakage caused by the high-capacity style encoder. In comparison, by adding style equalization, we successfully reduce content leakage and replicates style, as demonstrated by the CER and the style opinion score that are close to that of real handwriting samples.

## 5.4 ANALYSIS OF STYLE ATTENTION

Here we analyze how our model utilizes the time-dependent style information contained in the reference example $\mathbf{x}^r$ by examining the attention weights of the style attention module. As discussed in Sec. 4, while our style representation is a time-dependent sequence, style equalization can only transfer time-independent global style information from the style reference. Therefore, when we apply

---

[3]Due to privacy reasons, the handwriting reference examples shown in the paper and the supplemental material are synthetic. They are close reproductions of unseen real styles using a generative model with a different architecture. The generations shown here are very similar when real samples are used as style input. All the evaluations reported in Fig. 3d are done using real unseen style examples.

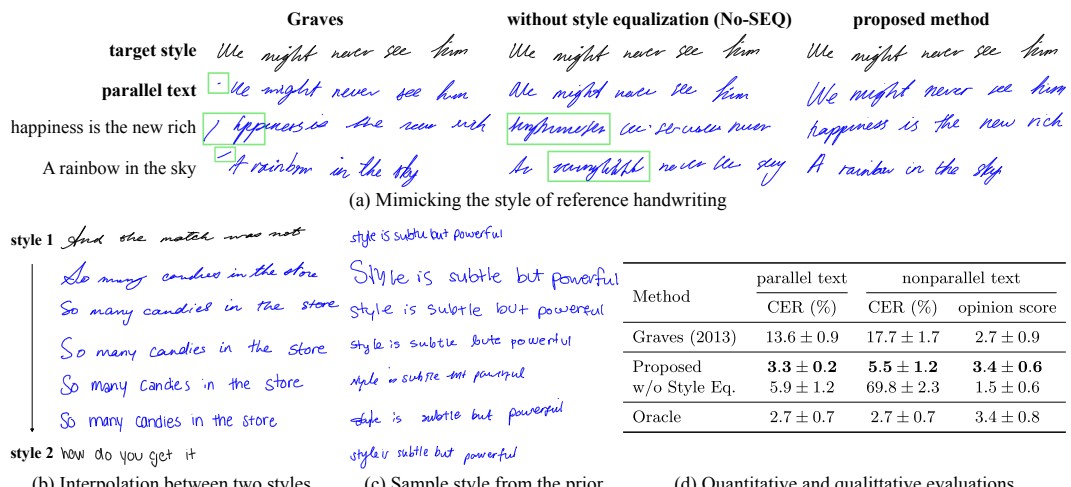

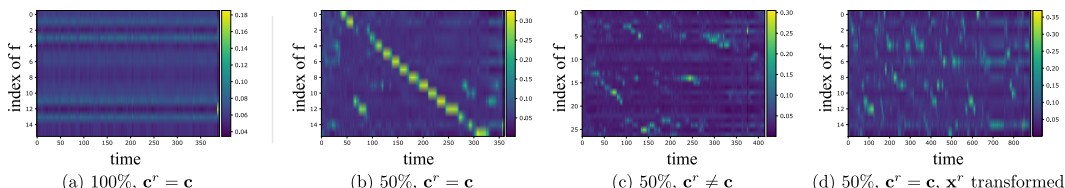

| Method | parallel text CER (%) | nonparallel text CER (%) | opinion score |
|---|---|---|---|
| Graves (2013) | $13.6 \pm 0.9$ | $17.7 \pm 1.7$ | $2.7 \pm 0.9$ |
| Proposed | $\mathbf{3.3 \pm 0.2}$ | $\mathbf{5.5 \pm 1.2}$ | $\mathbf{3.4 \pm 0.6}$ |
| w/o Style Eq. | $5.9 \pm 1.2$ | $69.8 \pm 2.3$ | $1.5 \pm 0.6$ |
| Oracle | $2.7 \pm 0.7$ | $2.7 \pm 0.7$ | $3.4 \pm 0.8$ |

(a) Mimicking the style of reference handwriting

(b) Interpolation between two styles   (c) Sample style from the prior   (d) Quantitative and qualittative evaluations

Figure 3: **Handwriting generation results and evaluation.** The reference style examples are shown in black, and the outputs are shown in blue.

Figure 4: **Style attention weights.** The figure shows the style attention weights of two models trained on LibriTTS during inference on unseen styles with (a) style equalization always applied during training (100%) and (b,c,d) style equalization applied to 50% of the training batches.

style equalization to all the samples during training, we learn a time-independent representation. This can be seen from Fig. 4a where the attention weights become constant over time during inference.

We encourage our style encoder to learn the time-dependent aspects of style by applying style equalization only to 50% of the training samples. As can be seen from the time-varying attention weights in Fig. 4b-d, this training procedure enables the style encoder to utilize style information more efficiently by focusing time instances with similar signal and context. For example, when the style reference contains the target content, *i.e.*, $\mathbf{c}^r = \mathbf{c}$ (Fig. 4b), the attention weights form a block-diagonal pattern, indicating the model focuses to the time instances of the style signal that match the current content and context. In Fig. 4c, we show attention weights when $\mathbf{c}^r \neq \mathbf{c}$, where the weights are still well localized over time to gather pieces of time-dependent style information from matching signal in $\mathbf{x}^r$. When we use style equalization to transform style during inference, Fig. 4d, the weights are more diffuse since it can only transfer global style.

The flexibility to apply and remove style equalization (and hence the representation bottleneck) during training is one of the main differences compared to existing methods (Wang et al., 2018; Hu et al., 2020; Hsu et al., 2018), which operate on ground-truth input, suffer from the training-inference mismatch, and thus require tight bottlenecks to the style representation. In contrast, with style equalization, we are able to use high-dimensional style (*i.e.*, large $s$ and $k$) and retain time-dependent style information.

## 6 CONCLUSION

This paper proposes a simple but effective method, style equalization, to learn a generative sequence model where style and content can be controlled separately. The generative model supports 1) accurate replication of styles from a single style reference, 2) interpolation between two reference styles, and 3) generating new reference styles. Experiments on speech and handwriting domains show that the method obtains state-of-the-art synthesis results.

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

## A  BROADER IMPACT

The technology we develop in the paper, like other artificial intelligence technologies, potentially has both positive and negative impacts to our society (Brundage et al., 2018). One potential risk associated with all generative models is creating fake digital content. If deployed irresponsibly, speech and handwriting synthesis could facilitate deceptive interactions, including mimicking a person's voice or handwriting in automatic speared phishing attacks, gaining security access, or directing public opinions. Examples of responsible deployments of the technology include (but not limited to):

- A system-level authentication every time the technology is used to register style or generate output.
- An encryption system to protect registered style.
- A watermarking system (for both speech and handwriting) such that the generations can be easily identified by human or detection systems (Hua et al., 2016).

Beyond system-level security measures, technologies to identify fake digital content have also been rapidly developed (Chen et al., 2020; Lyu, 2020; Güera and Delp, 2018; Dolhansky et al., 2020; Alegre et al., 2013; Wu et al., 2017; Evans et al., 2013; Kinnunen et al., 2020). Despite the potential negative societal impacts, we believe that the technology will have a larger positive impact, such as enabling new accessibility capabilities (*e.g.*, helping mute people speak in their voices and paralyzed people write in their handwriting) and better human-computer interaction (*e.g.*, by improving downstream speech and handwriting recognizers).

## B  DETAILS OF THE TRAINING PROCEDURE

Training with style equalization is straightforward — we jointly optimize all the model parameters by maximizing the log-likelihood lower bound as described in eq. (1) of the main paper. We found the following optional steps to be useful to improve training and the quality of the generation.

- During training, we found it helpful to add a small amount of Gaussian noise to the ground-truth $x_{t-1}$ that is fed back to the bottom LSTM. Intuitively, we are simulating the noise caused by sampling the output distribution during inference at training time. We found that it makes the inference more stable. A similar method is proposed by Meng et al. (2021) to learn auto-regressive models.
- To encourage the basis $A$ that is used by $\phi$ in eq. (2) in the main paper to capture a wide variety of styles, we maintain $A$ as an orthonormal basis. We normalize each column $a_i$ in $A$ to be unit norm in the architecture and minimize $|a_i^\top a_j|^2$ for all $i$ and $j \neq i$. The minimization is conducted by minimizing the trace of $(A^\top A)^2$, which is estimated efficiently using Hutchinson's trace estimator (Hutchinson, 1989) with 100 random samples from a standard Normal distribution. The estimated value is used as a regularization with loss weight equal to 1.

Overall, we optimize the following objective function

$$\max_{\theta, A} \mathop{\mathbb{E}}_{\substack{(\mathbf{x}, \mathbf{c}) \sim \mathcal{X} \\ (\mathbf{x}', \mathbf{c}') \sim \mathcal{X} \\ \mathbf{n} \sim \mathcal{N}(0, \sigma^2)}} \sum_{t=1}^{M} \mathop{\mathbb{E}}_{z_t \sim q_\theta(z_t | \mathcal{M}_\theta(\mathbf{x}', \phi_\theta(\mathbf{x}', \mathbf{x}), \mathbf{x}_{1..t-1}, \mathbf{c})} \log p_\theta(x_t | z_t, \mathbf{x}_{1..t-1} + \mathbf{n}_{1..t-1}, \mathbf{c})$$

$$- \mathcal{D}_{KL}\left(q_\theta(z_t | \mathcal{M}_\theta(\mathbf{x}', \phi_\theta(\mathbf{x}', \mathbf{x}), \mathbf{x}_{1..t-1}, \mathbf{c}) \,\|\, p_\theta(z_t | \mathbf{x}_{1..t-1}, \mathbf{c})\right) - \mathrm{tr}\left((A^\top A)^2\right), \quad (3)$$

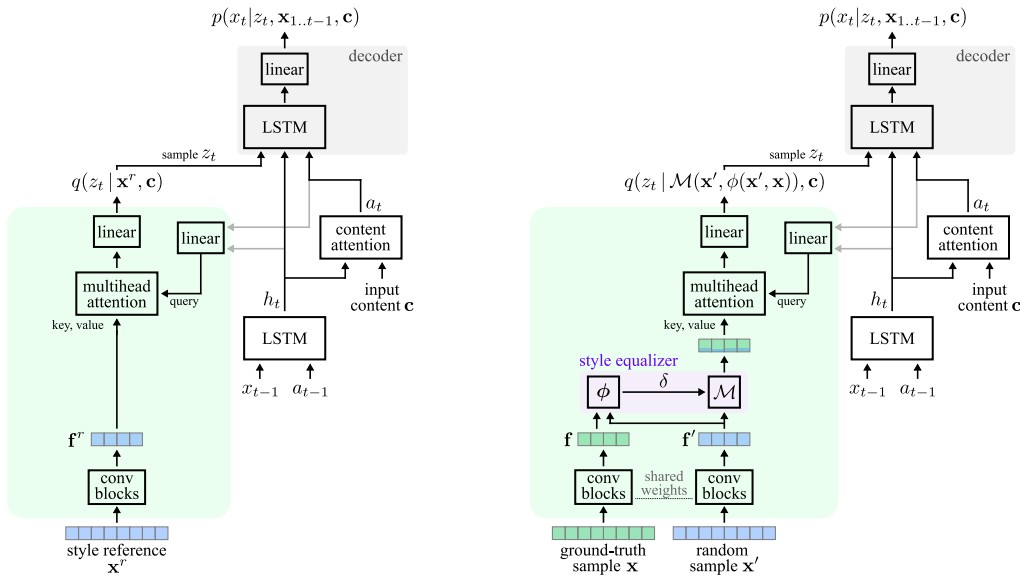

(a) overview of the entire model        (b) overview of the entire model with style equalization

Figure 5: **Overview of the model.** (a) shows an overview of the entire model without style equalization (used during inference since $\delta = \phi(\mathbf{x}^r, \mathbf{x}^r) = 0$). It includes a style encoder (in green), a content attention, a decoder (in gray), and a LSTM at the bottom. Note that the input content $\mathbf{c}$ can be the output of a content-embedding network (used in speech synthesis) or one-hot encoding of characters (used in handwriting synthesis). $a_t$ is the output of the content attention at time $t$, which is a linear combination of the elements in $\mathbf{c}$. (b) shows an overview of the entire model with style equalization (used during training or during interpolation). $\phi$ computes the vector $\delta$ that encodes the amount of style transformation between $\mathbf{x}'$ and $\mathbf{x}$. $\mathcal{M}$ applies this transformation to $\mathbf{x}'$ to match the style of $\mathbf{x}$. Please see Table 4 for details about individual blocks.

where $\theta$ represents all network parameters (except $A$). We use the reparameterization trick that is commonly used in variational autoencoders (Kingma and Welling, 2014) with one sample to estimate the inner expectation in eq. (3), and we use ADAM (Kingma and Ba, 2015) with $\beta_1 = 0.9$, $\beta_2 = 0.98$, and a learning rate schedule used by Vaswani et al. (2017) with a warm-up period of $4,000$ iterations to optimize the objective function. The learning rate increases rapidly within the warm-up period to $10^{-4}$ and decreases slowly after then.

## C  MODEL ARCHITECTURE DETAILS

In this section, we provide more details about the model architecture used in the paper. Fig. 5 shows an overview of the model, and Table 4 lists the individual block formulations used in Fig. 5. We use the same architecture for both handwriting and speech synthesis, except for the hyper-parameters, which we list at the end of the section.

The model can be separated into a backbone and a style encoder. The backbone is composed of a decoder, content attention, and an LSTM at the bottom. It is a standard model architecture and has been used and extended in many works of handwriting and speech synthesis (Graves, 2013; Shen et al., 2018; Wang et al., 2018; Hsu et al., 2018).

The bottom LSTM is one-layer, and it accumulates information from the past outputs $\mathbf{x}_{1..t-1}$ and the previously attended content $a_1, \ldots, a_{t-1}$ into its hidden state $h_t$. The content attention, which is proposed by Graves (2013), utilizes moving Gaussian windows to calculate attention weights for the content. The focused content $a_t$ at time $t$ is a linear combination of the elements in $\mathbf{c}$ based on the attention weights. The decoder at the top is a two-layer LSTM that takes all available information, including $h_t$, $z_t$, and $a_t$, and outputs the parameters of $p(x_t|z_t, \mathbf{x}_{1..t-1}, \mathbf{c})$. As mentioned in Sec. B, we add a small Gaussian noise to $x_{t-1}$ that is passed to the bottom LSTM.

Table 4: Block formulation used in Fig. 5

| Block name | Architecture |
| --- | --- |
| conv blocks | blur $\rightarrow$ conv$(3, f_1, 2, 0)$ $\rightarrow$ Swish $\rightarrow$ dropout$(0.1)$ $\rightarrow$ 
 blur $\rightarrow$ conv$(3, f_2, 2, 0)$ $\rightarrow$ Swish $\rightarrow$ dropout$(0.1)$ $\rightarrow$ 
 blur $\rightarrow$ conv$(3, f_3, 2, 0)$ $\rightarrow$ Swish $\rightarrow$ dropout$(0.1)$ $\rightarrow$ 
 blur $\rightarrow$ conv$(3, f_4, 2, 0)$ $\rightarrow$ Swish $\rightarrow$ dropout$(0.1)$ 
 $(f_1, f_2, f_3, f_4) = \begin{cases} (32, 64, 128, 256), & \text{for handwriting} \\ (256, 384, 512, 512), & \text{for speech} \end{cases}$ |
| multihead attention | number of heads = 4 

 dimension of query, key, value $= \begin{cases} 128, & \text{for handwriting} \\ 64, & \text{for VCTK} \\ 192, & \text{for LibriTTS} \end{cases}$ |
| bottom LSTM | number of layers = 1 
 dimension $= \begin{cases} 512, & \text{for handwriting} \\ 2048, & \text{for speech} \end{cases}$ |
| top LSTM | number of layers = 2 
 dimension $= \begin{cases} 512, & \text{for handwriting} \\ 2048, & \text{for speech} \end{cases}$ |
| content attention | number of Gaussian windows = 10 
 See Graves (2013) for the exact formulation. |
| content encoder (speech-only) | conv$(5, 256, 1, 2)$ $\rightarrow$ Swish $\rightarrow$ conv$(5, 256, 1, 2)$ $\rightarrow$ Swish $\rightarrow$ 
 conv$(5, 256, 1, 2)$ $\rightarrow$ Swish $\rightarrow$ bidirectional LSTM (dim=256) |

`blur`: 1D low-pass filtering with kernel $[1\ 3\ 3\ 1]$
`conv`$(k, f, s, p)$: 1D convolution with kernel size $k$, feature dimension $f$, stride $s$, and padding $p$
`Swish`: Swish nonlinearity Ramachandran et al. (2018); Hendrycks and Gimpel (2016)
`dropout`$(p)$: dropout with probability $p$

Our style encoder is composed of a 4-layer convolutional network and a multi-head attention layer. Given a style reference input $\mathbf{x}^r$, the convolutional network extracts the feature sequence $\mathbf{f}^r$. We apply low-pass filtering before every sub-sampling (Zhang, 2019) to avoid aliasing caused by sub-sampling in the convolutional network. Given the past outputs $h_t$ and the currently focused content $c_t$, we use multi-head attention to extract relevant information from $\mathbf{f}^r$. The query vector is computed from $h_t$ and $c_t$ using a linear layer, and the key and the value vectors are the individual feature vectors contained in the sequence $\mathbf{f}^r$ without positional encoding. The intuition is that if the model plans to write a specific character or utter a specific word, it should find the information in the style reference and mimic it. The variational approximation $q(z_t|\cdot)$ is a multivariate Gaussian distribution with a diagonal covariance matrix. The prior distribution $p(z_t|\mathbf{x}_{1..t-1}, \mathbf{c})$ is also modeled as a multivariate Gaussian distribution with a diagonal covariance matrix, and we a two-layer feed-forward network to compute its means and standard deviations from $h_t$ and $c_t$. When the style equalization is used, $\mathcal{M}$ and $\phi$ are inserted into the style encoder, as shown in Fig. 5(b).

Now we summarize the hyper-parameters used for handwriting and speech. Please also see Table 4.

- For handwriting, the dimension of all LSTMs are 512. The final linear layer outputs a 122-dimensional vector, which is used to parameterize the output distribution. The output distribution includes a mixture of 20 bivariate Gaussian distributions that model the pen movement, a Bernoulli distribution for pen lifting, and a Bernoulli distribution for sequence stops. The posterior and the prior Gaussian distributions are 256-dimensional. The convolutional network in the style encoder has four layers; all of them use kernel size 3, stride 2, and no padding. Their feature dimensions are $3 \rightarrow 32 \rightarrow 64 \rightarrow 128 \rightarrow 256$. We use dropout with a dropping rate equal to 0.1 after each nonlinearity in the convolutional network. The multihead attention has 4 heads, the dimension of

the query, key, and value vectors are all 256. The dimension of $\delta$ (*i.e.*, $k$) is 128. The input content **c** is represented as 200-dimensional one-hot vectors of the input characters. The standard deviation of the Gaussian noise is 0.1, and during inference, we reduce the standard deviation of the output distribution to 0.9 of the original one. The model is trained for 100 epochs on a machine with 8 A100 GPUs, and the training took 36 hours.

- For speech, all LSTMs have the same 2048 dimension. The final linear layer outputs a 484-dimensional vector, which is used to parameterize the output distribution. The output distribution includes a mixture of three 80-dimensional Gaussian distributions with diagonal covariance that models the mel-spectrogram and a Bernoulli distribution for sequence stops. The convolutional network in the style encoder has four layers; all of them use kernel size 3, stride 2, no padding. Their feature dimensions are $80 \rightarrow 256 \rightarrow 384 \rightarrow 512 \rightarrow 512$. We use dropout with a dropping rate equal to 0.1 after each nonlinearity in the convolutional network. The multihead attention has 4 heads, the dimension of the query, key, and value vectors are all 256.

  The input sentence is represented as phonemes, which contain 148 symbols. We follow the pre-processing used by Shen et al. (2018) for phoneme and mel-spectrogram extraction. We also follow Shen et al. (2018) and use a bidirectional LSTM and a convolutional network to encode the dependencies between phonemes. The architecture is the same as that used by Shen et al. (2018). The posterior and prior Gaussian distributions are 512-dimensional, and the dimension of $\delta$ (*i.e.*, $k$) is 64 for VCTK dataset and 192 for LibriTTS dataset. The standard deviation of the added noise is 0.2, and during inference, we reduce the standard deviation of the output distribution to 0.74 of the original one. The VCTK model is trained for 70 epochs on a machine with 8 A100 GPUs, and the training took 12 hours; the LibriTTS model is trained for 25 epochs on a machine with 8 A100 GPUs, and the training took 3 days.

## D    STYLE CLASSIFIER NETWORK

As we mentioned in Sec. 5.2, we train a speaker classifier using the objective function proposed by Deng et al. (2019). Using the features extracted by the speaker classifier, we measure the style similarity between two waveforms using the cosine similarity between the features. The style classifier comprises a convolutional network, an LSTM, and a linear layer that transforms the last hidden state into the feature that we use to compute the cosine similarity. The input to the convolutional network is the 80-dimensional phoneme, which is extracted using the same procedure as the one used by Shen et al. (2018). The convolutional network has four layers; all of them use kernel size 3, stride 2, valid padding, and swish non-linearity (Ramachandran et al., 2018). Their feature dimensions are $80 \rightarrow 256 \rightarrow 384 \rightarrow 512 \rightarrow 512$. We use dropout with a dropping rate equal to 0.1 after each non-linearity in the convolutional network. The LSTM has one, and its dimension is 512. We split the training set of LibriTTS-all-960 into the training, validation, and test sets by a ratio of 85%, 7.5%, 7.5%, respectively. We use the same learning rate schedule and optimizer mentioned above to train the classifier. The classifier achieves 96.5% validation accuracy.

## E    MORE SPEECH AND HANDWRITING GENERATION RESULTS

Our results can be viewed at `https://apple.github.io/ml-style-equalization`. In the website, we show an extensive list of speech and handwriting samples generated by the proposed and the baseline methods. Note that it may take a while for the speech results to load, and if the audio players do not contain the play button, please increase the size of the browser window. To remove the effect of the vocoder when comparing synthesized speech samples with real speech samples, all real speech samples (including those in the style opinion score evaluations) are converted to mel-spectrogram and reconstructed back to waveform using the same vocoder that is used by the generative models, *i.e.*, waveglow (Prenger et al., 2019).

For speech synthesis, the webpage contains

- a video showcasing the generation of speech with various styles and content
- nonparallel-text generation with seen speakers from LibriTTS-all-960
- nonparallel-text generation with unseen speakers from LibriTTS-all-960
- an ablation study that compares training with and without style equalization

- interpolation between two unseen style reference speech
- generated speech with random styles sampled from the learned prior
- nonparallel-text generation with seen speakers from VCTK
- parallel-text generation with seen speakers from VCTK

For handwriting synthesis, the webpage contains

- a video showcasing the online generation of handwriting with various styles and content
- nonparallel-text generation with unseen style
- parallel-text generation with unseen style
- generated handwriting with random styles sampled from the learned prior
- interpolation between two unseen style reference handwriting

## F  ADDITIONAL ABLATION STUDY: THE ROLE OF $\mathbf{x}'$

During training, the proposed style equalization randomly selects a sample from the training dataset  NEW
as $\mathbf{x}'$, which is unrelated to the ground-truth $\mathbf{x}$. One question is naturally raised: "Since $\mathbf{x}'$ is
unrelated to $\mathbf{x}$, do we really need it to be a sample?". Theoretically, since our design of $\mathcal{M}$ and $\phi$
in eq. (2) prevents content leakage and transfers time-independent ground-truth style information
from $\mathbf{x}$ through $\mathbf{z}$, as long as $\mathbf{x}'$ does not contain the content information about $\mathbf{x}$, the learned model
should be able to control style and content separately during inference. To verify the hypothesis, we
conduct the following ablation studies:

1. $\mathbf{x}'$ is a fixed vector: We initialize $\mathbf{x}'$ as a random vector but fixed it during training and inference.
2. $\mathbf{x}'$ is a random noise: We randomly sample $\mathbf{x}'$ from the standard Gaussian distribution.

As can be seen from Table 5, both methods are able to train models that can separately control content
and style (low WER and `avgRank` and high `cos-sim`). They achieve higher style replication
quality than GST-nS, which has additional speaker information. Nevertheless, the proposed model
(*i.e.*, using real samples as $\mathbf{x}'$) is able to utilize both time-independent and *time-dependent* style
information of $\mathbf{x}$ (see discussion in Sec. 5.4), and thus, our model still outperforms the two models.

There are additional disadvantages when using random noises as $\mathbf{x}'$ instead of real samples:

- During inference, the two models still need to run $\mathcal{M}$ and $\phi$. In comparison, our usage of $\mathbf{x}'$ allows us to remove $\mathcal{M}$ and $\phi$ when mimicking a reference example.
- Interpolation between two reference examples becomes non-trivial. By design, our usage of $\mathbf{x}'$ enables the model to learn to transform the style from one real sample to another. In contract, the other two models only learn to transfer style to a random noise (or a fixed vector). Thus, while interpolation is straight-forward for our model, it is not for the two methods.

Table 5: Ablation study results on the role of $\mathbf{x}'$. All models are trained on LibriTTS-all-960 dataset.

| Method | Seen speakers, parallel text | | | Seen speakers, nonparallel text | | |
|---|---|---|---|---|---|---|
| | WER (%) | cos-sim ↑ | avgRank ↓ | WER (%) | cos-sim ↑ | avgRank ↓ |
| $\mathbf{x}'$ is a fixed vector | $8.0 \pm 0.2$ | $0.69 \pm 0.22$ | $14 \pm 78$ | $7.1 \pm 0.1$ | $0.70 \pm 0.20$ | $12 \pm 95$ |
| $\mathbf{x}'$ is random noise | $8.4 \pm 0.0$ | $0.72 \pm 0.17$ | $4.7 \pm 32$ | $\mathbf{6.7 \pm 0.2}$ | $0.72 \pm 0.16$ | $5.1 \pm 56$ |
| Proposed ($\mathbf{x}'$ is a real sample) | $\mathbf{6.2 \pm 0.5}$ | $\mathbf{0.82 \pm 0.14}$ | $\mathbf{1.7 \pm 4.1}$ | $9.4 \pm 0.3$ | $\mathbf{0.78 \pm 0.14}$ | $\mathbf{1.8 \pm 6.0}$ |

| Method | Unseen speakers, parallel text | | | Unseen speakers, nonparallel text | | |
|---|---|---|---|---|---|---|
| | WER (%) | cos-sim ↑ | avgRank ↓ | WER (%) | cos-sim ↑ | avgRank ↓ |
| $\mathbf{x}'$ is a fixed vector | $9.2 \pm 0.3$ | $0.53 \pm 0.18$ | $22 \pm 97$ | $\mathbf{6.2 \pm 0.1}$ | $0.53 \pm 0.18$ | $23 \pm 103$ |
| $\mathbf{x}'$ is random noise | $8.9 \pm 0.0$ | $0.49 \pm 0.16$ | $14 \pm 55$ | $6.3 \pm 0.2$ | $0.48 \pm 0.16$ | $13 \pm 56$ |
| Proposed ($\mathbf{x}'$ is a real sample) | $\mathbf{6.8 \pm 0.1}$ | $\mathbf{0.63 \pm 0.15}$ | $9.0 \pm 63$ | $7.6 \pm 0.9$ | $\mathbf{0.57 \pm 0.15}$ | $\mathbf{7.4 \pm 43}$ |

# G Overview of related works

NEW

We provide a high-level summary of various controllable sequence generative models in Table 6. In the table, we compare the methods on their needs of : (1) user identification or pretrained embedding, (2) phoneme or character segmentation, (3) content recognizer or pretrained encoder, (4) their training loss and procedure, and (5) the applications shown in the papers. As can be seen and to the best of our knowledge, while there exist many controllable sequence generative models, our proposed method is the first that does not require user ID, segmentation, pretrained recognizer and proven to be applicable on both speech and handwriting domains.

Table 6: Overview of controllable sequence models. The table provides a high-level overview of various controllable sequence models. For details, please see individual references.

| Method | No user ID or embedding needed | No segmentation needed | No recognizer needed | Training method | Domain |
|---|---|---|---|---|---|
| (Kim et al., 2020) | ○ | ● | ● | log-likelihood | speech |
| (Chen et al., 2021) | ○ | ○ | ● | log-likelihood | speech |
| (Donahue et al., 2020) | ○ | ● | ● | adversarial | speech |
| (Donahue et al., 2020) | ○ | ● | ● | adversarial | speech |
| (Kameoka et al., 2020) | ○ | ● | ● | log-likelihood | speech |
| (Gibiansky et al., 2017) | ○ | ◗ (pretrained) | ● | log-likelihood | speech |
| (Jia et al., 2018) | ○ | ● | ● | log-likelihood | speech |
| (Kaneko et al., 2019b) | ○ | ● | ● | adversarial | speech |
| (Kaneko et al., 2019a) | ○ (group data) | ● | ● | adversarial | speech |
| (Kaneko and Kameoka, 2018) | ○ (group data) | ● | ● | adversarial | speech |
| (Kameoka et al., 2018) | ○ | ● | ● | adversarial | speech |
| (Davis et al., 2020) | ● | ◗ (pretrained) | ○ | adversarial + log-likelihood | handwriting |
| (Kang et al., 2020) | ○ | ● | ○ | adversarial + log-likelihood | handwriting |
| (Bhunia et al., 2021) | ○ | ● | ○ | adversarial + log-likelihood | handwriting |
| (Kotani et al., 2020) | ○ | ◗ (pretrained) | ● | log-likelihood | handwriting |
| (Hsu et al., 2018) | ◗ (not on LibriTTS) | ● | ● | log-likelihood | speech |
| (Hu et al., 2020) | ● | ● | ◗ (pretrained content encoder) | adversarial + log-likelihood | speech |
| (Aksan et al., 2018) | ● | ○ | ○ | log-likelihood | handwriting |
| (Akuzawa et al., 2018) | ● | ● | ● | log-likelihood + KL-annealing | speech |
| (Henter et al., 2018) | ● | ○ | ● | log-likelihood | speech |
| (Sun et al., 2020) | ○ | ○ | ● | log-likelihood | speech |
| (Ma et al., 2018) | ● | ● | ● | adversarial | speech |
| (Graves, 2013) | ● | ● | ● | log-likelihood | handwriting |
| GST (Wang et al., 2018) | ● | ● | ● | log-likelihood | speech |
| Proposed style equalization | ● | ● | ● | log-likelihood | speech, handwriting |

# H Utilizing time-dependent style information

NEW

In the section, we answer the question: "does our model utilize time-dependent style information?" When a model fully utilizes time-dependent style information, in the parallel setting (where the input text appears exactly in the reference example), the generated output should be a reconstruction of the reference example. On the other hand, if a model utilizes only time-independent style information, its outputs in the parallel setting can still be different from the reference example, *i.e.*, the same sentence spoken by the same user can be very different in different contexts. The difference in style replication quality provides a basis to the following evaluation.

We first evaluate the capability of our model to utilize time-dependent style information by comparing the results between parallel (where all the time-dependent style information is available) and non-parallel (where little time-dependent and mostly time-independent style is available) settings. As can be seen in Table 1, Table 2 and Fig. 3d, our results (WER, CER, and cos-sim) in the parallel setting are better than those in the non-parallel setting. The results show that the model is capable of utilizing the time-dependent information in the reference example.

Next, we verify this capability by visualizing the style attention weights in the three scenarios in Fig. 4b-d. In the parallel setting (Fig. 4b), the model strongly attends the correct time instance; in the non-parallel setting (Fig. 4c), the model attends to localized pieces in the reference example; in

the transformed setting (Fig. 4d), the style attention weights are more diffused over the sequence. Additionally, we qualitatively evaluate the capability via the handwriting/speech generation results provided in the supplementary material. As can be seen, the reproduction is "qualitatively closer" in the parallel setting and in the nonparallel setting when a character appears in the style reference within a similar context, *e.g.*, in Fig. 6b, the character 'A' and 'a' in the first non-parallel text example and 'p' in the second example. We also provide a nonparallel example that has overlapping content as the reference style in the third example in Fig. 6b. As can be seen, the generated result is qualitatively similar to the reference example at the overlapped regions even when they are separated by unrelated text.

The ablation study in Table 5 compares our model (which is trained to utilize time-dependent style information via applying style equalization to 50% of the batches) with two models that always transform style inputs (and thus can only use time-independent style information). While all models in the comparison use the same style encoder, our model, with its capability to utilize time-dependent style information, achieves the highest cosine similarities. The style attention weights shown in Fig. 4a and Fig. 4b also verify that our training method enables the use of time-dependent information.

Finally, we compare the same model trained with our method (which applies style equalization to 50% of the batches) and entirely without style equalization in Fig. 3d. While both models can utilize time-dependent information, the model trained without style equalization suffers from content leakage (as shown by the high CER in the nonparallel text setting). In comparison, the proposed model is less effected by the settings.

In summary, the proposed method (utilizing the style attention module and applying style equalization on half of the batches) enables the model to utilize time-dependent style information while avoiding catastrophic content leakage.

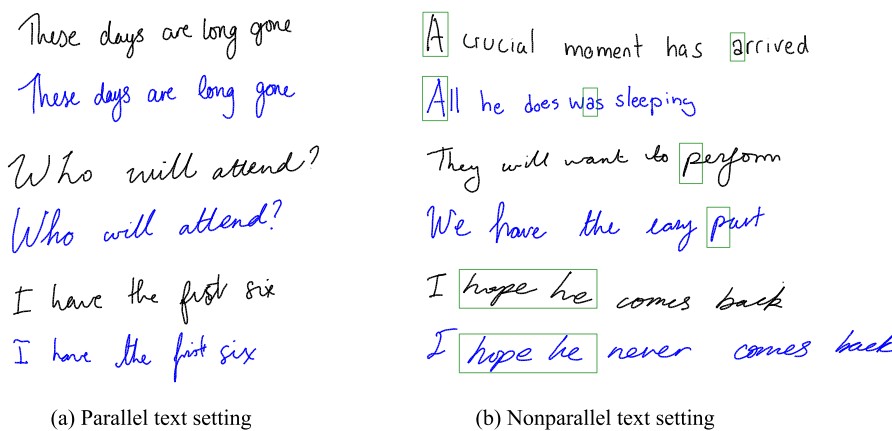

(a) Parallel text setting        (b) Nonparallel text setting

Figure 6: Parallel and nonparallel text generation results. In the figure, the reference examples are shown in black, and the generated results are shown in blue.

