# OpenReview forum: "Style Equalization: Unsupervised Learning of Controllable Generative Sequence Models "
_ICLR.cc/2022/Conference — ICLR 2022 Submitted_

### Official Review · Reviewer_A5yD · 2021-10-24

**Correctness:** 2
**Technical Novelty And Significance:** 3
**Empirical Novelty And Significance:** 2
**Recommendation:** 3
**Confidence:** 4

**Main Review:**

Strengths:
 - Proposed an interesting method with nice experimental results.

Weaknesses:
 - The connection to related works is not well discussed.
 - The benefit of the proposed method over other similar methods is not well justified.

Detailed comments:
 1. The main idea of this paper is based on VRNN (Chung et al., 2015). Such connection, as well as the connection to other methods based on VRNN (e.g. Aksan et al., 2018), is not clearly discussed in this paper. Other major approaches on style transferring are not well discussed either, such as VAE-based approaches (Akuzawa et al., 2018, Henter et al, 2018, Sun et al., 2020).
 2. This paper claims "state-of-the-art". However, the experiments are not compared to the state-of-the-art approaches in the field.
    - The TTS experiments focus on speaker similarity, but is compared to the Global Style Tokens (GST) model, which is not good at voice modeling due to its limited dictionary. VAE-based approach, such as Hsu et al., 2019, are better baselines. For the Tacotron baseline, it's also worth to include the result conditioned on speaker ID.
    - On handwriting synthesis, the baseline used in the paper, Graves, 2013, is not a strong baseline either. More recent works, such as Aksan et al., 2018, Kotani et al., 2020, are better baselines.
 3. The main novelty of this paper is Eq (2). The idea is: to generate sample $x$, the model is conditioned on both a content $c$ and a style $z$, where $z$ is from a function $M(x, x')$. Here $x'$ is a sample unrelated to $x$. A natural question is: is $x'$ really used by the model, or is it actually ignored? This is neither theoretically nor empirically answered in the paper.
 4. To address the above question empirically, I'd suggest running two ablation studies:
    - replace $M(x, x’)$ with $M(x)$
    - replace $x’$ (more precisely $f'$) with a learned prior
    - (interesting but not required) replace $x’$ (more precisely $f'$) with a random noise
 5. There are quite some redundancy in the statement of the mismatch problem. The key idea of this paper is not presented until paper 5. It would be nice to save some space from them, and use it for the connection to related works.

References:
 - Chung et al., A Recurrent Latent Variable Model for Sequential Data, NeurIPS 2015.
 - Aksan et al., DeepWriting: Making Digital Ink Editable via Deep Generative Modeling, 2018.
 - Akuzawa et al., Expressive Speech Synthesis via Modeling Expressions with Variational Autoencoder, Interspeech 2018.
 - Henter et al., Deep Encoder-Decoder Models for Unsupervised Learning of Controllable Speech Synthesis, 2018.
 - Sun et al., Fully-hierarchical fine-grained prosody modeling for interpretable speech synthesis, ICASSP 2020.
 - Hsu et al., Hierarchical Generative Modeling for Controllable Speech Synthesis, ICLR 2019.
 - Kotani et al., Generating Handwriting via Decoupled Style Descriptors, ECCV 2020.

**Summary Of The Paper:**

 - Proposed an unsupervised style transferring framework based on VRNN;
 - Conducted experiments on TTS and handwriting synthesis to demonstrate the effectiveness of the proposed method.

**Summary Of The Review:**

While this paper proposes an interesting idea for style transferring with nice experimental results, the benefit of the proposed idea over reasonable baselines is not clearly exhibited. The connection to related works is also not well discussed.

---

> ### Author Response · Authors · 2021-11-17
> **Reply to the questions from Reviewer A5yD from the authors (2/2)**
>
> (continues "Reply to the questions from Reviewer A5yD from the authors (1/2)")
>
> **[3] It's also worth to include the result of Tacotron conditioned on speaker ID.**
>
> Thank you for the suggestion. We have conducted the experiments and included the results of Tacotron conditioned on speaker ID (Tacotron-S) on both VCTK (Table 1) and LibriTTS (Table 2) in the revised paper.  The results are as expected, while conditioning on speaker ID improves style replication (higher cos-sim and lower avgRank) than the vanilla Tacotron, our models have much better style replication capability.
>
> **[4] A natural question is: Is $x’$ really used by the model, or is it actually ignored? This is neither theoretically nor empirically answered in the paper.**
>
> This is a good question. Our model does not ignore $x’$, and it is empirically verified by our results.  Please see General Reply 1 above for our detailed answer.
>
>
>
> **[5] To address the above question empirically, I'd suggest running two ablation studies:**
>
> 1. *replace $M(x, x′)$ with $M(x)$*
> 2. *replace $x’$ with a learned prior $x′$*
> 3. *(interesting but not required) replace $x’$ with a random noise*
>
>
> Thank you for the suggestions.  We have conducted the experiments and included the results in the revised paper (Appendix F).
>
> From our understanding, (1: replacing $M(x,x’)$ with $M(x)$) and (2: replacing $x’$ with learned parameters) are similar to using a fixed noise as $x’$ for all $x$, since in all these cases, the fixed vector (or learned $x’$) simply becomes part of the model parameters.  Therefore, we conducted two experiments on LibriTTS:
>
> * replace $x’$ with a fixed vector
> * replace $x’$ with a Gaussian noise (randomly sampled every time)
>
> In theory, since both methods still utilize the $M$ and $\phi$, which transfer only the time-independent information of $x$, both methods should work and have minimal content leakage. However, there are a few drawbacks associated with the methods:
>
> 1. During inference, we still need to run $M$ and $\phi$.  In comparison, our usage of $x’$ allows us to remove $M$ and $\phi$ during inference.
> 2. The models trained by the two methods cannot utilize time-dependent style information even when they are available. This is because for the two methods, during both training and inference, $x$ is processed by $M$ and $\phi$, which remove time-dependent style information. In comparison, our model can utilize time-dependent style information (as discussed in Section 5.4).  As a result, it achieves higher cos-sim and lower spk-rank in the new Table 5 than the two methods.
> 3. Interpolation becomes non-trivial. By design, our usage of $x’$ enables the model to learn to transform the style from one sample to another during training. Thus, as shown in the paper, interpolation is easy for our models.  Using the two methods loses this benefit.
>
> We have included the discussion and results in Appendix F.  As can be seen, while the two models outperform GST, our model still achieves the best style replication quality.
>
> We would also like to clarify that even though utilizing a fixed vector or random noise as $x’$ produces good models, it does not mean that models trained with the proposed method ($x’$ is a sample from the training set) would ignore $x’$, as we have empirically shown in the paper and answered above.
>
>
> **[6] There are quite some redundancy...use it for the related works.**
>
> Thank you for the suggestion.  We have revised the paper and discussed the suggested reference.

---

> > ### Comment · Reviewer_A5yD · 2021-11-20
> > **Reply**
> >
> > Thank the authors for the responses and especially for running the additional experiments. They are very helpful for understanding this work.
> >
> > I still have a few major concerns remaining:
> >
> > 1) Strong claims not supported by the experiments.
> >
> > This paper claims the state-of-the-art. However, as the authors agreed, there are better baseline models existing that are not compared to in this work. Importantly, the baseline models chosen in the experiments are known to be weak on the specific tasks been evaluated (e.g. voice transferring for TTS).
> >
> > Even though, in the revised version, the claims are made even stronger, claiming "the first to demonstrate near-real content accuracy and style reproduction". It's not quite clear what this means or how it is justified. The experimental results don't fully support the claim, too (WER 9.5% from proposed, vs 6.6% from oracle in Table 1, similar in Table 2).
> >
> > 2) The experiment design is weak, both in terms of the baselines compared to, as well as the task/metrics being evaluated.
> >
> > Both the paper and the author responses emphasize on the capacity of time-dependent style modeling. However, all evaluations in the paper use time-independent metrics.
> >
> > For speech synthesis, evaluating the controlling of prosody instead of voice would be more proper.
> >
> > Also as mentioned, the baseline choices are weak.
> >
> > 3) The connection to other works.
> >
> > I would expect Sec 3 explicitly states that this model is built on top of VRNN, instead of referring to that work in an over-general way as "a variational lower bound of the likelihood".
> >
> > 4) The revision disclosed the affiliation of this work in footnote 1. This is not taken into account in my ratings, but the authors should be very careful on such disclosure.

---

> > > ### Author Response · Authors · 2021-11-21
> > > **Reply from the authors**
> > >
> > > Thank you for the reply and evaluation.  We have revised the paper according to your comments and added a new discussion in Appendix H.  We answer your questions below.
> > >
> > > **[1] Strong claims not supported by the experiments... Importantly, the baseline models chosen in the experiments are known to be weak on the specific tasks been evaluated (e.g. voice transferring for TTS). Even though, in the revised version, the claims are made even stronger, claiming "the first to demonstrate near-real content accuracy and style reproduction". It's not quite clear what this means or how it is justified. The experimental results don't fully support the claim, too (WER 9.5% from proposed, vs 6.6% from oracle in Table 1, similar in Table 2).**
> > >
> > > Thank you for the suggestion. We compared with the available state-of-the-art methods that can operate in the general setup discussed in the paper. For the other recent methods, we provided a detailed discussion and a comparison of capabilities (Section 2 and Appendix G). Please note that, except a few papers (Hsu et al. 2020 and Ma et al. 2018, note that code is not available for these methods), the other recent methods (Aksan et al. 2018) and (Kotani et al. 2020) cannot operate in the setup that our method operates (e.g. requiring user IDs or segmentation of phonemes/characters). We will be happy to include more if there is any suggestion. We have modified our discussion in the paper in Section 2 and Appendix G to reflect this.
> > >
> > > Thank you for pointing out to a potential source of confusion. We have toned down the claim and clarified the sentence as: “On LibriTTS, style equalization achieves close style replication (3.5 real oracle vs. 3.5 proposed in style opinion score) and content reproduction errors (6.6% real oracle vs. 9.5% proposed) to real samples.”
> > >
> > > **2. The experiment design is weak, both in terms of the baselines compared to, as well as the task/metrics being evaluated. Both the paper and the author responses emphasize on the capacity of time-dependent style modeling. However, all evaluations in the paper use time-independent metrics.  For speech synthesis, evaluating the controlling of prosody instead of voice would be more proper.**
> > >
> > > We reply to the comment regarding baselines in (1).
> > >
> > > We'd like to note that our main contribution is to tackle the "training-inference mismatch". Alleviating the problem enables us to use a sequence representation for style.
> > >
> > > We evaluate “the capability to model time-dependent style of speech/handwriting signals” by comparing the results between parallel (where all the time-dependent style information is available), non-parallel (little time-dependent and mostly time-independent style is available), and transformed (only time-independent style information is available) settings.  As can be seen in Table 1, Table 2 and Figure 3d, Style Equalization results (WER, cos-sim) in the parallel setting improve upon those in the non-parallel setting, due to the model’s capability to transfer the time-dependent style.
> > >
> > > We verify this capability by visualizing the style attention weights in the three scenarios in Figure 4b-d.  In the parallel setting (Figure 4b), the model strongly attends to the correct time instance; in the non-parallel setting (Figure 4c), the model attends to localized pieces in the reference example; in the transformed setting (Figure 4d), the style attention weights are more diffused over the sequence.
> > >
> > > Additionally, we qualitatively evaluate the capability via the handwriting/speech generation results provided in  Figure 6 in the revised paper (and in the supplementary material).  As can be seen, the reproduction is “qualitatively closer“ in the parallel setting and in the nonparallel setting when a character appears in the style reference within a similar context, e.g., the character ‘A’ and ‘a’ in the first non-parallel text example and ‘p’ in the second example.  We also provide a nonparallel text example that has overlapping content as the reference style in the third example.  As can be seen, the generated result is qualitatively similar to the reference example at the overlapped regions even when they are separated by unrelated text.
> > >
> > > Finally, the new ablation study in Appendix G compares our model with two models that always transform style input (and thus can only utilize time-independent style information).  While all models in Table 5 use the same style encoder, our model, with its capability to utilize time-dependent style information, achieves the highest cosine similarities.
> > >
> > > We have revised the paper and discussed this in Appendix H.
> > >
> > > **3. I would expect Sec 3 explicitly states that this model is built on top of VRNN, instead of referring to that work in an over-general way as "a variational lower bound of the likelihood".**
> > >
> > > Thank you for the suggestion. We have revised the paper as: “Specifically, we maximize a variational lower bound of the likelihood as VRNN (Chung et al., 2015)”.

---

> ### Author Response · Authors · 2021-11-17
> **Reply to the questions from Reviewer A5yD from the authors (1/2)**
>
> Thank you for your careful evaluation and valuable suggestions. We have answered each of your five questions in two replies.
>
> **[1] Connections between VRNN/VAE-based methods (Chung et al., 2015; Aksan et al., 2018; Akuzawa et al., 2018, Henter et al, 2018, Sun et al., 2020).**
>
> Thank you for suggesting the related (Akuzawa et al., 2018, Henter et al., 2018, Sun et al., 2020).  We have added them to Section 2 in the revised paper.  The followings are detailed discussions between these methods and ours:
>
> The connection between (Chung et al. 2018):
>
> You are correct. The proposed method (eq 1) optimizes the variational likelihood lower bound proposed in (Chung et al., 2015), and thus our model belongs to the category of VRNN.
>
> The connection between (Aksan et al. 2018):
>
> As discovered in many follow-up works of (Chung et al. 2018), utilizing the variational bound does not provide separated controls over content and style. Thus in (Aksan et al., 2018), the style and characters are separately modeled by two encoders.  However, the method needs to know the ground-truth character to write at every time step. As a result, the method requires ground-truth character segmentation during training and a handwriting recognizer during inference. Since the segmentation map on cursive handwriting is ill-defined, the generation of cursive handwriting can suffer from the use of character segmentation. The method also separately models character and word boundaries. During inference, the method utilizes priming to acquire an initial hidden state that contains the style information of the reference example and performs character recognition to determine character boundaries.  In comparison, our proposed method directly learns to align the input characters and output handwriting. We do not need ground-truth character segmentation during training nor running handwriting recognition during inference.  We also do not utilize priming.
>
> The connection between (Akuzawa et al., 2018; Henter et al., 2018; Sun et al., 2020):
>
> Thank you for suggesting the related works (Akuzawa et al., 2018; Henter et al., 2018; Sun et al., 2020).  We have discussed them in the revised paper. These works also utilize the variational lower bound and focus on learning style without style annotations.  To alleviate content leakage caused by the training-inference mismatch, Akuzawa et al. (2018) use KL cost annealing, which starts with a large KL loss weight and gradually decreases.  Henter et al. (2018) reveal an interesting connection between VQ-VAE and approximate maximum likelihood. They focus on learning the emotion of speech, but their method requires phoneme segmentation (i.e., forced alignment).  Sun et al. (2020) focus on learning the prosody of words and phonemes; however, on multi-speaker datasets like VCTK, the method requires pretrained speaker embedding and phoneme segmentation.  While these methods and our proposed method all use the variational lower bound, our method does not need forced alignment, speaker embedding, or careful tuning of the KL divergence loss weights.
>
> We have discussed these related works in the revised paper (Section 2 and Appendix G). If required, we are happy to provide the detailed discussion to the paper.
>
> **[2] The experiments are not compared to the state-of-the-art approaches in the field, such as (Hsu et al., 2019; Aksan et al., 2018; Kotani et al., 2020).**
>
> Please see our General Reply 2 above, where we answer the question.
>
> We agree that (Hsu et al., 2019; Aksan et al., 2018; Kotani et al., 2020) are state-of-the-art methods, as we discussed in the paper.  However, we cannot compare with these methods because they operate under different settings than ours or lack publicly available implementation (please see the following table).
>
> In the paper, we compare with Tacotron, Tacotron with speaker embedding (as suggested), GST (Wang et al. 2018), GST with speaker embedding, and (Graves, 2013). They are strong baselines, have widely-used publicly-available implementations, and are well understood by the community.  We evaluate them thoroughly with quantitative metrics (Word Error Rate and cosine similarity) and user study and provide numerous examples in the supplemental website. Thus, we believe our experiments provide valuable information to the community.
>
> | | do not use user ID or embedding | do not use character segmentation | public implementation |
> |-----|:----:|:----:|:----:|
> | Hsu et al. 2019 | no (on LibriTTS) | yes | no |
> | Aksan et al. 2018 | yes | no | yes |
> | Kotani et al. 2020 | no | yes (train a recognizer) | no |
> | GST (Wang et al. 2018) | yes | yes | yes |
> | Graves, 2013 | yes | yes | yes |
> | Ours | yes | yes | ongoing |

---

### Official Review · Reviewer_E4io · 2021-11-02

**Correctness:** 3
**Technical Novelty And Significance:** 2
**Empirical Novelty And Significance:** 3
**Recommendation:** 6
**Confidence:** 4

**Main Review:**

Strengths:
Controllable sequential generative models have been studies for years. One of the most fundamental problem is how to effectively capture the content information and style information, respectively. It is a critical while very challenging research problem, because the 'content' and 'style' are entangled in the training samples, and ones must carefully design the training objective such that each of these factor can be learned in a controllable way. The idea of learning the 'style equalization' is interesting, and achieves promising results on tasks in different application scenarios, i.e. TTS and text-to-handwriting synthesis. Despite that, the paper is easy to follow, and the demos show in the project page qualitatively demonstrate the proposed approach.

Weakness:
1.My main concern is the training of 'style equalization' module. Based on the design, the training is to approximate the posterior p(z|x, c) with q(z|M(x', phi(x, x)), c). In this such a design, the expectation is that the style encoder only capture the style information from the given samples x and x'. However, it might be possible that the network is trained to find a shortcut that conv(x) -> x_content+x_style, and M & phi are just trained to diminish x'. In this way, the training loss is still very small, however, the conv(.) does not trained to encode pure style information as expected.  I am wondering, is there any penalties or training tricks are used to avoid such shortcuts? As I do not find implementation details, and either ablations or analysis on the latent space that inferred by the style encoder conv(.), so I am not fully convinced.
2. To deal with the 'training-inference mismatch' issue, there are another line of works hat encourage disentanglement of 'content'
 and 'style'. For example, in [1], the model are also trained with paired and un-paired text-speech by disengaging the content and style information in each encoders. While, I do not see comparisons and discussions about this line of work. I am very curious to see what are the differences when applying 'disentanglement' and 'style equalization'.
3. There lack of implementation and experimental details.
4. The comparing SOTA are quite a few. On TTS, the model is only compared with Tacotron-based models, e.g. GST and other versions. On text-handwriting, is only Graves.
5. There lack of comprehensive ablation studies. Especially on the style latent space. I am curious to see differences of the latent space directly inferred by the style encoder conv(.), and the after by applying M( ., phi(.)).
6. No source code is submitted, it raises doubles on the reproducible ability.

[1] S. Ma, D. McDuff, Y. Song. NEURAL TTS STYLIZATION WITH ADVERSARIAL AND COLLABORATIVE GAMES

**Summary Of The Paper:**

In this paper, the authors argue that the typical training algorithms for controllable sequence generative models suffers from the 'training-inference mismatch'. Therefore, to address such a problem, they introduce a style transformation module that is called 'style equalization'. Such a module is designed to enable training using different content and style samples and thereby mitigate the training-inference mismatch problem. To demonstrate the generality of the proposed approach, the 'style equalization' is applied to two tasks of TTS and text-to-handwriting synthesis on three datasets. On both tasks. the models show good results.

Controllable sequential generative models have been studies for years. One of the most fundamental problem is how to effectively capture the content information and style information, respectively. It is a critical while very challenging research problem, because the 'content' and 'style' are entangled in the training samples, and ones must carefully design the training objective such that each of these factor can be learned in a controllable way. The idea of learning the 'style equalization' is interesting, and achieves promising results on tasks in different application scenarios, i.e. TTS and text-to-handwriting synthesis. Despite that, the paper is easy to follow, and the demos show in the project page qualitatively demonstrate the proposed approach.

**Summary Of The Review:**

This paper is well motivated, the idea of applying 'style equalization' is interesting. Also, the showed experimental results are impressive.

Upon the weakness I've pointed, I would suggest the authors:
1) demonstrate how M & phi can be trained in the expected way, which avoids the model collapse to shortcuts.
2) add discussions and comparisons with the very related line of work, i.e. content-style disentanglement
3) adding ablation studies and comparisons with more SOTA approaches

Minors:
By looking into the paper, I am not clear about the training and evaluation datasets. For example, on TTS, what is the training dataset and what is the evaluation dataset? Are they trained on VoxCeleb, and then evaluated on VCTK & LibirTTS. Or they trained on VCTK and evaluated on VCTK, and the same for LibriTTS?  If the latter, are the evaluation performed on 'seen' or 'unseen' speakers?

---

> ### Author Response · Authors · 2021-11-17
> **Reply to the questions from Reviewer E4io from the authors**
>
> Thank you for your careful evaluation and valuable suggestions. We have answered each of your questions in the following.
>
> **[1] “demonstrate how $M$ and $\phi$ can be trained in the expected way, which avoids the model collapse to shortcut” and “it might be possible that the network is trained to find a shortcut that conv(x) → x_content+x_style, and $M$ & $\phi$ are just trained to diminish $x’$“**
>
> Thanks for asking the question. Please see our General Reply 1 above, where we answer the question. Style equalization is designed to avoid forming the shortcut, and our results in the paper verify it.
>
>
> **[2] “add discussions and comparisons with content-style disentanglement” and “another line of works that encourages disentanglement of 'content' and 'style'. For example, in [Ma et al.], the model are also trained with* *paired and un-paired text-speech by disengaging the content and style information in each encoders.“**
>
> Thank you for suggesting the related work. (Ma et al., 2018) uses both parallel and nonparallel text-audio pairs to train the model. Intuitively, it detects whether the content in the generated output is correct and whether the generated output is “real” enough using a discriminator. Three kinds of losses (adversarial, style, and log-likelihood/reconstruction losses) are used to train the model.  While both methods (Ma et al. and Style Equalization) focus on learning to mimic style in an unsupervised manner, our method directly optimizes the log-likelihood, which significantly simplifies the training procedure. We have discussed (Ma et al., 2018) in the related work section.
>
>
> **[3] “lack of implementation and experimental details”**
>
> We provide these details in Appendix B, C, D. We understand the importance of reproducibility and intend to publish the code upon acceptance of the paper.
>
>
> **[4] “not clear about the training and evaluation datasets.  Are they trained on VoxCeleb, and then evaluated on VCTK & LibirTTS. Or they trained on VCTK and evaluated on VCTK, and the same for LibriTTS? If the latter, are the evaluation performed on 'seen' or 'unseen' speakers?”**
>
> Sorry for the confusion.  The models were trained on VCTK and evaluated on VCTK and the same for LibriTTS. All models in Table 1 are trained on VCTK, and all models in Table 2 are trained on LibriTTS. Only GST-nS uses additional training data from VoxCeleb. All results on VCTK uses seen speakers, and we tested both seen and unseen speakers for LibriTTS, indicated in Table 2.  We have revised the paper and clarified it (Section 5.2).
>
>
> **[5] “add ablation study”**
>
> We conduct the following ablation studies in the paper:
>
> * We have added a new ablation study in the revised paper (Appendix G), where we study the role of $x’$ by using random noise or a fixed vector as $x’$ during training, as suggested by Reviewer A5yD.
> * Our interpolation results (Figure 3b, supplemental website) show the outputs before and after $M$ and $\phi$ are applied. When we interpolate from style 1 to style 2, $\alpha$ goes from 0 to 1 (please see Section 4 for the definition of $\alpha$). Thus, the interpolation results demonstrate how the outputs look like when we gradually increase the effect of style equalization (from none to full).  Additionally, Figure 4b and Figure 4d show the style attention weights before ($\alpha=0$) and after ($\alpha=1$) applying style equalization, respectively. Specifically, $x^r$ is kept the same in Figure 4b and 4d, and Figure 4d shows the attention weights after we apply style equalization to $x^r$.
> * The table (Figure 3d) shows a quantitative comparison between our handwriting model trained with and without style equalization.
> * Figure 3a shows a qualitative comparison between our handwriting model trained with and without style equalization.
> * On the supplemental website, we show  a qualitative comparison between our speech model trained with and without style equalization on LibriTTS.
> * In Figure 4, we show the style attention weights of two of our models trained with and without using our training method discussed in Section 5.4.  The figure also shows the style attention weights of our model when the input contains parallel texts or nonparallel texts.
> * Example outputs generated with random samples from the latent space (Figure 3c, supplemental website) also provide valuable information about the latent space.
>
> We believe these ablation studies (together with the new study in Appendix G) provide valuable information for the community to understand and build insights into our method.
>
>
> **[6] “add comparison with SOTA”**
>
> Please see our General Reply 2 above, where we answer the question.  We have expanded the discussion and added a comparison table in Appendix G.

---

> > ### Comment · Reviewer_E4io · 2021-11-19
> > **Thanks for your response**
> >
> > Thanks for your detailed responses, it clarifies my main concerns.
> > Overall, I think the authors present an interesting idea, and the results demonstrate the effectiveness of the proposed approach.
> > So I would like to remain my original rating which leans to an 'accept'.

---

### Official Review · Reviewer_K825 · 2021-11-03

**Correctness:** 4
**Technical Novelty And Significance:** 3
**Empirical Novelty And Significance:** 4
**Recommendation:** 8
**Confidence:** 4

**Main Review:**

Strong points

- The problem this paper tackles is crucial and practical in controllable generations. In particular, the unsupervised learning and non-parallel setting are realistic. Also, the proposed method and the motivation behind it are simple but effective. The method is also technically sound.
- Overall, this paper is neat, clearly written, and well-organized.
- The experimental results are much improved in the automated evaluations and human evaluation. In particular, I was hard to distinguish the synthesized speech examples whether it is ground-truth or generated. The low WER of generated speech is very impressive for me.
- The authors provide sufficient experimental details for reproducibility and mentioned several training techniques.

Weak points

- Please clarify that, in Sec. 5.2, which data are compared in the 'cos-sim' metric?
- The implementation codes are not submitted. Although the authors mentioned the details of training, publish the source codes would be helpful for the community.

Questions

- Please refer to the weak points above.

**Summary Of The Paper:**

- To enhance the quality of style-controlled generation, especially in an unsupervised manner and non-parallel setting, this paper proposes a "style equalization" mechanism to prevent the content leakage problem. In the style equalization module, the style of a sample is transformed to be the same as the style of ground truth. The authors assumed that content information is time-dependent whereas the style can be time-independent so that the authors employ time-average pooling to learn the global style. Then the style difference is added to the inputs style features. At each time step, a content attended feature queries and attends appropriate style equalized feature via the multihead attention module. The entire model is optimized to maximize the ELBO. The proposed method is demonstrated on speech synthesis and hand-writing synthesis tasks.

**Summary Of The Review:**

Recommendation

- I vote for "accept" with the reasons that the proposed method is simple and somewhat novel, and also it shows strongly effective results. I think the results are able to contribute to the style-controlled synthesis communities.

---

> ### Author Response · Authors · 2021-11-17
> **Reply to the questions from Reviewer K825 from the authors**
>
> Thank you for your careful evaluation and valuable suggestions. We have answered each of your questions in the following.
>
> **[1] “In Sec. 5.2, which data are compared in the 'cos-sim' metric?”**
>
> The cosine similarity (cos-sim) is computed between each reference example and the corresponding generated result.  We have added the information to the paper (Section 5.2).
>
> **[2] “publish the source codes would be helpful for the community.”**
>
> We completely agree with the comment. We intend to publish the code upon acceptance of the paper.

---

> > ### Comment · Reviewer_K825 · 2021-11-29
> > **Thanks for your responses**
> >
> > Thanks for clarifying the metric and deciding the publication of source codes. With the interesting idea 'style equalization, I would like to remain my original rating 'accept'.

---

### Author Response · Authors · 2021-11-17
**General Reply from the authors (2/2)**


(continue General Reply from the authors (1/2))

**2. Comparison with other state-of-the-art controllable sequence models, e.g., (Hsu et al., 2019; Aksan et al., 2018; Kotani et al., 2020).**

In the revised paper, we have added Appendix G and Table 6, which give an overview of related controllable sequence models.  Below, we give a simplified version of the table listing the most related works to ours.


|                        | do not use user ID or embedding | do not use character segmentation | public implementation |
|------------------------|:-------------------------------:|:---------------------------------:|:---------------------:|
| Hsu et al. 2019 | no (on LibriTTS) | yes | no |
| Ma et al. 2018 | yes | yes | no |
| Aksan et al. 2018      |               yes               |                 no                |          yes          |
| Kotani et al. 2020     |                no               |      yes (train a recognizer)     |           no          |
| GST (Wang et al. 2018) |               yes               |                yes                |          yes          |
| Graves, 2013           |               yes               |                yes                |          yes          |
| Ours                   |               yes               |                yes                |        ongoing        |


As can be seen from the table, we operate in a different setting than (Aksan et al. 2018) and (Kotani et al. 2020).  Due to the lack of publicly available implementation of (Hsu et al. 2019) and (Ma et al. 2018), we cannot directly compare with them. Nevertheless, we have discussed the differences in the revised paper (Section 2). We also want to point out that while (Hsu et al., 2019) supports training without speaker IDs on a small proprietary dataset, in their experiments on the larger LibriTTS dataset, they utilized speaker IDs. In comparison, our proposed method does not use speaker/writer IDs at all on VCTK, LibriTTS, and the handwriting datasets.

In the paper, we compare with Tacotron, Tacotron with speaker embedding (as suggested by Reviewer A5yD), Global Style Token (Wang et al. 2018), Global Style Token with speaker embedding, and (Graves, 2013). They are strong baselines, have publicly-available implementations, and are well understood by the community. We evaluate them thoroughly with quantitative metrics (Word Error Rate and cosine similarity) and user study. We also provide numerous examples in the supplemental website. Thus, we believe our experiments provide valuable information to the community.

Unlike discriminative models that can be compared using a public benchmark, evaluation of generative models require subjective tests or specialized evaluation (e.g. style replication accuracy) that could only be done by running the source code. The results of the existing methods are highly dependent on the implementation details, therefore without an implementation from the authors or a good public implementation accepted by the community, it’s difficult to provide a fair comparison. We try our best effort to provide a strong comparison with the best methods available within these constraints. We also provide an alternative view based on the capabilities and discuss these methods in detail (please see the table, and more details and discussions in the paper). We appreciate the reviewers’ evaluation based on these constraints and would be happy to include additional discussions/evaluation if they suggest more.


We have conducted the requested experiments and answered to individual questions from each reviewer in separate comments. Please let us know if there is any further question.

---

### Author Response · Authors · 2021-11-17
**General Reply from the authors (1/2)**

We thank all reviewers for their careful and valuable evaluation and for pointing out the strengths of our work:

* Great results (All reviewers:  K825: “strongly effective results”, E4IO: “results are impressive”, A5yD: “nice experimental results”)
* Focus on an important problem (K825, E4io)
* Proposed method is simple but effective (K825) and interesting (E4io, A5yD)
* Paper is well-organized and easy to follow (K825, E4io)


There are two common questions (separately answered here and in "General Reply from the authors (2/2)"):

**1. In Figure 2, is the randomly-drawn $x’$ from the training set utilized by the model, or is it actually ignored and the model directly uses the ground truth $x$ (the shortcut)?**

Our model does not ignore $x’$, and it is verified by our results:

1. During inference, when mimicking the style of a reference example, we input the example through $x’$ and remove the style equalizer (since $\phi(x, x)=0$ by design). If the model ignored $x’$, we would not be able to generate results shown in Figure 3, the supplemental website, and the quantitative evaluation in Tables 1-3 and Figure 3d.
2. The interpolation results shown in Figure 3b and the supplemental website require style information from both $x$ and $x’$ (Section 4). For example, in Figure 3b, the model utilizes information from $x’$ to mimic style 1, and it smoothly transitions to use information from $x$ when mimicking style 2.
3. The attention weights in Figure 4b show that the model attends to the most relevant parts of $x’$ during the generation process.

The following design choices in style equalization help avoid the shortcut:

1. Intuitively, when the style of $x$ and $x’$ are similar, $\phi(x, x’)$ will be small. This means that little information is passed from $x$ to $x’$, so the model has to utilize $x’$, which contains unrelated content but relevant style. This capability is achieved by $\phi$ subtracting $f$ and $f’$ in a linear subspace of $A$ (eq 2).
2. To encourage the model to utilize time-dependent style information when available, for half of the training batches, we set $x = x’$, or equivalently removing $M$ and $\phi$ and directly use $f’$.  Please see Section 5.4 in the paper for more discussion.

We have conducted an ablation study suggested by Reviewer A5yD. We have added the results in Appendix F. The results show the effectiveness of the proposed method and the design choices.

---

### Decision · Program_Chairs · 2022-01-20

**Decision:**

Reject

**Comment:**

This work aims to improve style transfer in the unsupervised non-parallel case. It does this by proposing a style equalization approach to prevent content leakage and assuming that content information is time-dependent whereas style information is time-dependent. This is an important problem to solve and lots of prior work in the area exists. The work is well-organised with good experimental results. However, there are strong claims in the paper and there is insufficient experimental comparison to similar related work such as Hsu et al. 2019 and Ma et al. 2018 to back that up. If there's no comparison with the current state of the art (e.g. due to a private implementation or dataset) then it's hard to justify calling a new work a new state of the art. Even though an implementation may be private, it can be worth spending time to reproduce a paper or asking the authors for an implementation. Finally task and metric selection could be improved to better highlight the performance of the approach. The reviewers thank the authors for the rebuttal but it was insufficient to change their decision.